# Fast raster-scan optoacoustic mesoscopy enables assessment of human melanoma microvasculature in vivo

Hailong He[1,2], Christine Schönmann[3], Mathias Schwarz[4], Benedikt Hindelang [3], Andrei Berezhnoi[1,2], Susanne Annette Steimle-Grauer[3], Ulf Darsow[3], Juan Aguirre[1,2] & Vasilis Ntziachristos [1,2✉]

Melanoma is associated with angiogenesis and vascular changes that may extend through the entire skin depth. Three-dimensional imaging of vascular characteristics in skin lesions could therefore allow diagnostic insights not available by conventional visual inspection. Raster-scan optoacoustic mesoscopy (RSOM) images microvasculature through the entire skin depth with resolutions of tens of micrometers; however, current RSOM implementations are too slow to overcome the strong breathing motions on the upper torso where melanoma lesions commonly occur. To enable high-resolution imaging of melanoma vasculature in humans, we accelerate RSOM scanning using an illumination scheme that is coaxial with a high-sensitivity ultrasound detector path, yielding 15 s single-breath-hold scans that minimize motion artifacts. We apply this Fast RSOM to image 10 melanomas and 10 benign nevi in vivo, showing marked differences between malignant and benign lesions, supporting the possibility to use biomarkers extracted from RSOM imaging of vasculature for lesion characterization to improve diagnostics.

[1] Institute of Biological and Medical Imaging, Helmholtz Zentrum München, Neuherberg, Germany. [2] Chair of Biological Imaging at the Central Institute for Translational Cancer Research (TranslaTUM), School of Medicine, Technical University of Munich, Munich, Germany. [3] Department of Dermatology and Allergy, Technical University of Munich, Munich, Germany. [4] iThera Medical GmbH, Munich, Germany. ✉email: bioimaging.translatum@tum.de

Cutaneous melanoma is one of most aggressive and fatal forms of skin malignancies, responsible for over 10,000 deaths annually in the United States alone[1–3]. Early diagnosis followed by rapid and complete surgical excision are essential for improving prognosis[2,4,5]. Currently, the diagnosis of melanoma is largely based on clinical assessment using dermoscopy and histological analysis of the excised lesion or biopsy[6–8]. Dermoscopy only provides a two-dimensional superficial assessment of a lesion, which does not extend beyond the papillary dermis due to photon diffusion [1,2]. Furthermore, visual examination is subjective and accuracy depends largely on the experience of the dermatologist[9]. These factors lead to many false positives and unnecessary biopsies, which are invasive, slow, and costly[5,10–12]. For example, it has been reported that the number of benign pigmented lesions excised to detect one melanoma varies from 6.3 to 8.7 for dermatologists and from 20 to 30 for general practitioners[13–16].

Malignant melanoma alters the local skin microvasculature in a manner different from benign lesions, with malignant lesions exhibiting higher vessel density[17–20]. As such, noninvasive imaging of morphological features of lesion microvasculature may complement superficial visual inspection for melanoma detection[19–22]. Furthermore, histological studies have reported that vascular changes induced by malignant melanoma can extend through the entire depth of the skin. It was found that tumor vascularity increases significantly with mean tumor thickness from 1–4 mm[18,20,22,23], while vessel patterns change such that thin melanomas have more homogeneous vasculature and thick melanomas exhibit more chaotic and heterogeneous vasculature[17]. However, current non-invasive imaging methods cannot fully resolve microvasculature throughout the entire depth of the skin, which hinders the full exploitation of tumor vascular features as biomarkers for melanoma detection.

For example, Doppler ultrasound can detect neovascularization of skin tumor lesions with reportedly high specificity for malignancy (90%–100%) but variable sensitivity (34–100%)[24]. However, Doppler ultrasound only resolves vessels with diameters > 100 μm in lesions with a thickness of >2 mm without contrast agents (microbubbles), resulting in poor sensitivity for early-stage melanoma detection[24,25]. Dynamic optical coherence tomography (D-OCT) based on speckle variance allows in vivo evaluation of skin vascular patterns[26] and has been applied to distinguish melanoma from benign lesions by examining the lesions' vasculature[17,27]. However, D-OCT can only reliably visualize vascular patterns of thin lesions up to a maximum depth of about 500 μm, mainly due to scattering[17,28]. This limited depth penetration misses changes in the deep dermal vasculature for melanoma detection.

In contrast, raster-scan optoacoustic mesoscopy (RSOM) images microvasculature at resolutions in the tens of microns through the whole skin depth by combining optical excitation with ultra-broad bandwidth ultrasound transducers[29–33]. Several studies highlight the potential of RSOM to assess tumor angiogenesis in animals and visualize the microvasculature of human skin, in vivo, using only endogenous contrast[30,31,34–38]. For example, RSOM can image the internal and surrounding vasculature of a melanoma tumor in a mouse, in vivo, at depths up to 3 mm[31]. RSOM has also been applied to visualize skin morphology and vascular patterns in the dermis and sub-dermis of psoriasis patients, enabling computation of several biomarkers to differentiate between psoriasis and healthy skin[30,32]. However, current RSOM systems can only scan a $4 \times 2$ mm$^2$ area in 70 s due to the maximum repetition rate of the laser (500 MHz), which is determined by the maximum permissible exposure of human skin to laser light[30,39]. This long scanning time makes RSOM susceptible to artifacts due to breathing or other motions. Motion correction algorithms reduce artifacts, but are insufficient to compensate for the large displacements that occur at the back and chest during breathing[40,41]. This weakness is problematic for imaging nevi or melanoma, which are frequently found on a person's upper torso, particularly the upper back.

We introduce here a fast RSOM (FRSOM) system based on a high-sensitivity ultrasound detector and a top coaxial illumination scheme, implemented by transmitting light using a single fiber through a central aperture of the high-sensitivity ultrasound transducer. These hardware innovations enable the use of a lower energy laser source at a high-repetition rate, affording an almost four-fold decrease in scan time, without sacrificing image quality or exceeding laser safety limits. FRSOM can scan a $4 \times 2$ mm$^2$ field-of-view in only 15 s, which is short enough for a patient to hold his or her breath during imaging of the back and chest. We apply single-breath-hold FRSOM to visualize the microvasculature of human melanoma and surrounding skin tissue at unprecedented resolution-to-depth ratios. We demonstrate that biomarkers such as vascular density, complexity, and tortuosity can be extracted from single-breath-hold FRSOM images and used to distinguish groups of cutaneous malignant melanomas and benign nevi. This study introduces FRSOM vasculature imaging as a noninvasive method to provide complementary information to dermoscopy and increase the accuracy of melanoma detection.

## Results

**FRSOM system.** The FRSOM system combines a custom-made spherically focused through-hole transducer with a coaxial illumination scheme (Fig. 1a, see also Methods), achieved by inserting a single fiber through the central aperture of the transducer (Supplementary Fig. 1). Compared to conventional RSOM, the coaxial illumination transmitted through a single fiber allows a 4-fold reduction in pulse energy and a concomitant increase in the laser repetition rate to 1.4 kHz (from 500 Hz), without decreasing the light fluence or exceeding laser safety limits (see Methods). In addition, an integrated miniature preamplifier was connected directly to the sensor circuits in the through-hole transducer, which minimized artifacts and doubled the FRSOM system's achievable SNR compared to the transducer used in conventional RSOM (see Methods and Supplementary Fig. 1). This combination of a high-repetition laser with a more sensitive transducer allows FRSOM to scan an area of $4 \times 2$ mm$^2$ in 15 s, with image quality comparable to that of conventional RSOM images recorded in 70 s. The full comparisons between the FRSOM and conventional RSOM were conducted by measuring the same skin tissue of a healthy volunteer and corresponding results are shown in Supplementary Fig. 1 and a volumetric FRSOM image is shown in Supplementary Movie 1.

To demonstrate the imaging performance of FRSOM, the microvasculature network of a volunteer's cuticle was measured; corresponding coronal and cross-sectional MIP (maximum intensity projection) images are shown in Fig. 1b, c. The capillaries (white arrows in Fig. 1b, c) in these images are clearly resolved with comparable image quality to that of conventional RSOM[36]. The FRSOM images were reconstructed over two frequency bands (10–40 MHz and 40–120 MHz; see image reconstruction in Methods). Larger structures are revealed in the lower frequency band and smaller structures in the higher frequency band. The two reconstructed bands are color-coded and overlaid in the rendered images (red: larger structures; green: smaller structures) so that finer vasculature is highlighted in the presence of larger vessels[42,43]. In addition, a multispectral FRSOM implementation was devised using a high-repetition-rate laser (1.4 kHz) with low pulse energy (see Methods). This

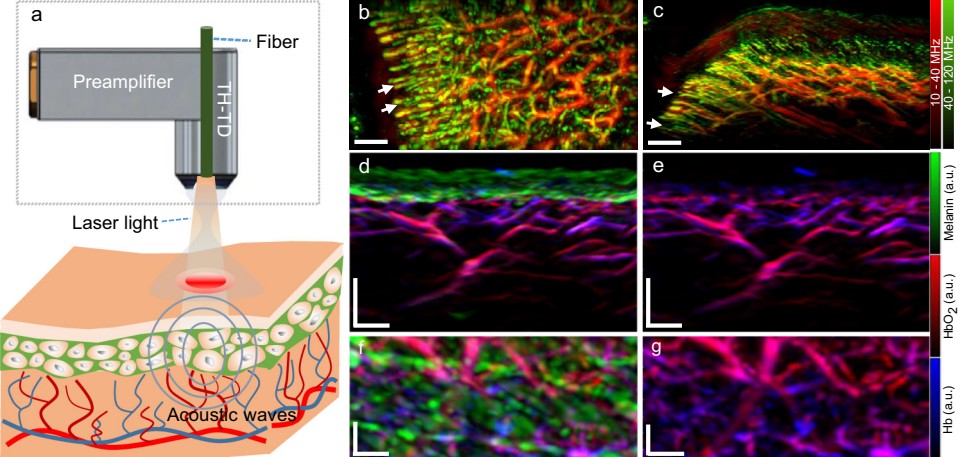

**Fig. 1 Schematic of the FRSOM system. a** Schematic of the FRSOM setup. The illumination scheme is implemented by transmitting light through the center aperture of the through-hole transducer (TH-TD) using a single fiber, which generates an illumination spot of 1.4 mm in diameter. **b**, **c** MIP (maximum intensity projection) images in the cross-sectional and coronal direction of the microvasculature networks from the cuticle of a healthy volunteer (male); White arrows indicate the capillaries. The images are color-coded to represent the two reconstructed frequency bands (red: larger structures in the bandwidth of 10–40 MHz; green: smaller structures in the bandwidth of 40–120 MHz). **d**, **f** MIP images in the cross-sectional and coronal direction of healthy skin structures recorded by multispectral FRSOM, where the reconstructed images were unmixed for melanin (green), oxyhemoglobin (HbO$_2$, red) and deoxyhemoglobin (Hb, blue). **e**, **g** MIP images of deoxyhemoglobin (blue) and oxyhemoglobin (red) with the melanin channel was switched off, allowing the missing melanin features to be easily identified in the superficial skin layer compared with the images of (**d**) and (**f**). The volunteer was measured multiple times with similar results. All scale bar: 500 μm.

multispectral FRSOM records a $4 \times 2$ mm$^2$ section of healthy skin with four wavelengths (532, 555, 579, and 606 nm) in 60 s, compared to 13 min for a conventional multispectral RSOM setup[38]. The reconstructed images were unmixed to highlight melanin (green), oxyhemoglobin (red), and deoxyhemoglobin (blue, Fig. 1d–g), achieving comparable unmixing performance to our conventional multispectral RSOM[38]. To visualize the location of melanin, the green channel (in which melanin absorbs) was switched off in the images of Fig. 1e, g, allowing the missing melanin features to be easily identified in the superficial skin layer compared with the images of Fig. 1d, f. Since the 60 s measurement time of this multispectral FRSOM implementation exceeds a single-breath-hold, a 25 s dual-wavelength (515 nm and 532 nm) version (see Methods) was also tested and shown to discriminate the melanin and hemoglobin distributions of skin tissue (see Supplementary Fig. S2).

**Efficacy of motion suppression in single-breath-hold FRSOM imaging**. Having established the imaging performance of FRSOM within a 15 s window, we sought to determine if a single-breath-hold by the patient during this time sufficed to suppress motion artifacts. To this end, we measured the healthy skin on the back of a female volunteer during both normal breathing and a 15 s single-breath-hold (Fig. 2). Figure 2a, b shows the raw optoacoustic signals of the FRSOM scans recorded with and without breathing motion, with the dashed lines indicating the skin surface. The recorded FRSOM data reveals the severe motion (maximum of 350 μm, beyond the capability of motion correction algorithms) caused by breathing (Fig. 2c), as well as the significant reduction in this motion (maximum of 70 μm) when the breath is held for 15 s (Fig. 2d, details of motion computation are described in the Methods). Figure 2e, f shows the corresponding cross-sectional MIP images with and without breathing. Breathing motion causes extreme blurring of vascular structures (Fig. 2e), which cannot be corrected by any motion correction algorithms, whereas the volunteer simply holding her breath for 15 s enables the collection of high-resolution images of the skin vasculature (Fig. 2f).

The performance of FRSOM was assessed further by imaging a psoriasis lesion and the surrounding healthy skin (Fig. 2g–l) on the same patient's back during 15 s single-breath-holds. The features of psoriatic skin at the human arm area were previously characterized using RSOM and provide a useful benchmark for the capabilities of FRSOM[30]. Figure 2g reveals that FRSOM can resolve and quantify known features of psoriatic skin, including elongated and dilated capillary loops (visualized in green) extending through the rete ridges almost to the skin's surface. Coronal FRSOM images close to the surface of the psoriatic skin (Fig. 2i) show the ends of the capillary loops, which appear markedly different from those in the corresponding image of healthy skin (Fig. 2j). In addition, the dermal vessels of the psoriatic skin (Fig. 2k) have larger diameters and appear denser than in the healthy skin (Fig. 2l). Overall, the previously documented epidermal thickening, capillary elongation, and increased dermal vascularization were clearly visualized by FRSOM during the 15 s breath-hold period.

**FRSOM vascular imaging of melanoma and surrounding skin tissue**. We thus far demonstrated that FRSOM resolves morphological and vascular features of healthy and psoriatic skin on a patient's back, during a single-breath-hold, with performance similar to that of conventional RSOM when applied to measurements on a human arm where motion is minimal. To further elucidate the suitability of FRSOM for clinical applications, the system was used to visualize changes in the skin microvasculature that had been altered by the progression of malignant melanoma. A melanoma lesion from a patient's back was scanned during 15 s breath-holds in three regions (Fig. 3a, scans 1–3 performed along the directions of the arrows), each with a field view of $4 \times 2$ mm$^2$ (red rectangle): the lesion base (scan 1,), the lesion edge (scan 2), and the surrounding skin tissue (scan 3). A histological image from the lesion base (scan 1) is shown in Fig. 3b. Cross-sectional FRSOM images and the corresponding MIP images in the coronal direction of the epidermis (EP) and dermis (DR) layers are shown in Fig. 3c–k. Volumetric reconstructions of the three scans are presented in Supplementary Movies S2–S4.

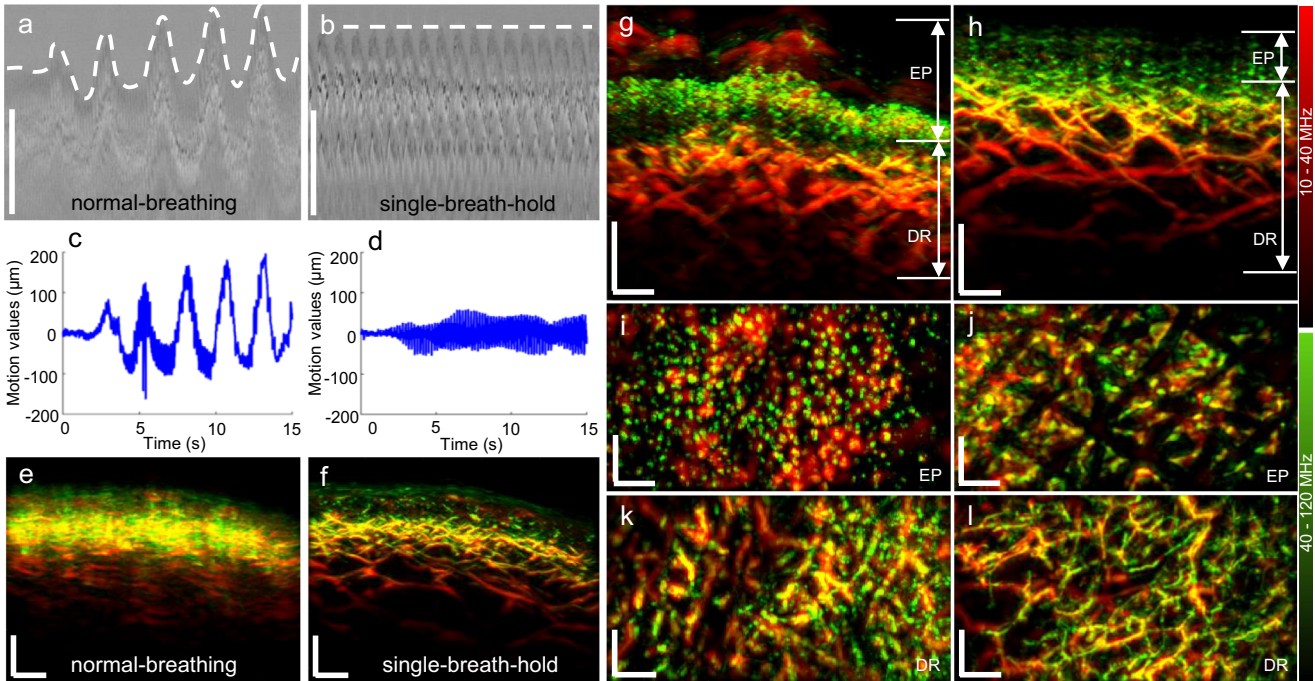

**Fig. 2 Single-breath-hold FRSOM imaging of healthy and psoriatic skin. a, b** Raw optoacoustic signals from FRSOM measurements of healthy skin on the back of a healthy volunteer (female), acquired during (**a**) normal breathing and (**b**) while the breath was held for 15 s (single-breath-hold). Dashed lines mark the positions of the skin surface in all A-line optoacoustic signals, which reflect the breathing motion. The x-axis represents the alignment of all A-line optoacoustic signals recorded in the 15 s from a fixed position. **c, d** Corresponding motion graphs of (**a**) and (**b**), which show significantly different motion patterns between normal breathing and the held breath. **e, f** Cross-sectional MIP images of healthy skin on the back of the healthy volunteer, acquired during (**e**) normal breathing and (**f**) while the breath was held for 15 s (single-breath-hold). A marked improvement in image quality is achieved when the breath is held during scanning compared to when normal breathing motions occur. **g, h** Cross-sectional MIP images of psoriatic skin and adjacent healthy skin from a patient's back (male). **i, j** Corresponding MIP images in the coronal direction of the epidermis (EP) layers of (**g**) and (**h**). **k, l** MIP images in the coronal direction of the vascular networks in the dermis (DR) layer of (**g**) and (**h**). All images are color-coded to represent the two reconstructed frequency bands (red: larger structures in the bandwidth of 10–40 MHz; green: smaller structures in the bandwidth of 40–120 MHz). The healthy volunteer and patient with psoriasis were measured multiple times with similar results. All scale bar: 500 μm.

To illustrate how FRSOM resolves the vascular characteristics of the melanoma lesion, we compared the skin features in the epidermal and dermal layers at the three scan positions. The cross-section images (Fig. 3c–e) reveal the layered structure of the skin and the strong contrast of melanin in the EP layer, which clearly delineates it from the DR layer. Compared with the surrounding skin tissue, dermal vessels of the lesion appear irregular and disordered. Figure 3f–h show coronal images corresponding to the EP layer in Fig. 3c–e. The coronal image of the lesion base (Fig. 3f) reveals the inhomogeneous melanin distribution within the melanoma. The boundary between the pigmented area and the adjacent tissue is clearly visible in the coronal image of the lesion's edge (Fig. 3g, white dashed line). The healthy tissue surrounding the lesion is characterized by less contrast from melanin in the coronal MIP image (Fig. 3h) than in the lesion's base and edge. Since angiogenesis is an important factor for tumor identification, we further investigated how the microvasculature features differed between the pigmented lesion areas and the surrounding skin tissue. Figure 3i–k shows coronal images corresponding to the DR layer of Fig. 3c–e. The dermal vasculature of the melanoma lesion and its edge exhibited a dense and dotted vessel pattern (Fig. 3i, j), characteristic of the disease[18,44]. In contrast, entire vessels are discernible in the surrounding unaffected skin tissue (Fig. 3k). In the image of lesion's edge (Fig. 3j), vascular patterns distinguish the pigmented lesion from the adjacent tissue (white dashed line), showing a transition of the vascular distribution from the highly dense and disordered pattern typical of melanoma to a regular vessel

network. In addition, the microvasculature of the skin bordering the lesion (Fig. 3j) presented with irregular, dotted, and comma-like structures compared to the regular vascular network of skin farther from the lesion (Fig. 3k). Moreover, the overall vessel density decreases from the pigmented area of the lesion (Fig. 3i) to the lesion's edge (Fig. 3j) to the tissue farther from the lesion (Fig. 3k).

**Quantification of FRSOM vasculature features between melanoma and dysplastic nevus.** Quantifiable biomarkers can be extracted from RSOM's high-resolution vascular images to provide objective measures of disease severity[30]. Here, we sought to evaluate this capability of FRSOM by quantifying and comparing pathophysiological features of benign dysplastic nevi and melanomas. For this, ten dysplastic nevi and ten melanomas were scanned on the upper torsos of ten patients by single-breath-hold FRSOM. Some lesions were measured at two different areas of the lesion's edge, to afford 16 datasets for each of the nevus and melanoma groups. A photograph of a dysplastic nevus located on a patient's back is shown in Fig. 4a; the red rectangle indicates the scanned region-of-interest at the edge of the lesion. A histological image and the corresponding cross-sectional FRSOM image are shown in Fig. 4b, c, respectively. A volumetric reconstruction of a nevus lesion is presented in Supplementary Movie S5. Figure 4f shows an exemplary melanoma lesion and a red rectangle indicating the scanned region. The corresponding histology and cross-sectional FRSOM image are depicted in Fig. 4g, h,

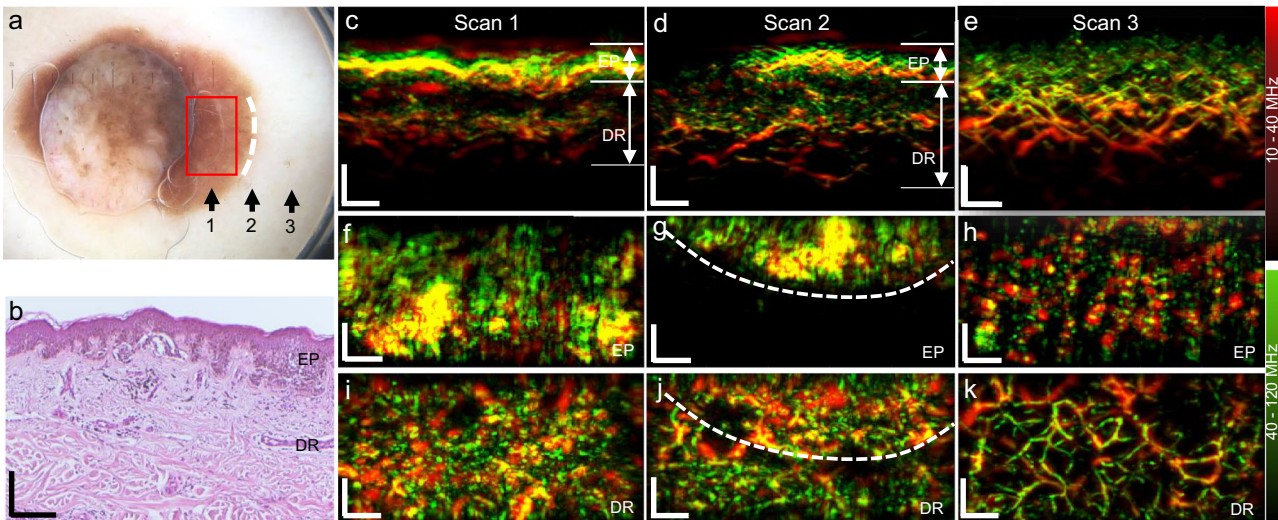

**Fig. 3 Vascular imaging of melanoma and the surrounding skin tissue from a patient with melanoma (female). a** Photograph of a melanoma lesion; Black arrows and labels (1, 2, 3) indicate the three scanning positions, corresponding to the tumor base, the lesion edge and the surrounding skin tissue. The red rectangle depicts the field-of-view of 4 × 2 mm². **b** Histological image of the melanoma sample corresponding to the area (label 1) marked by the red rectangle in (**a**). **c–e** MIP cross-sectional FRSOM images of the three scanned regions (labels 1, 2, 3). **f–h** MIP images in the coronal direction corresponding to the epidermal (EP) layer of (**c–e**). **i–k** Coronal images corresponding to the dermal (DR) layer of (**c–e**). The white dashed lines in (**g**) indicate the border between the pigmented lesion and the close surrounding skin tissue. Comparing the dermal vascular structures of (**i–k**), a dense dotted vascular pattern is clearly visualized in the tumor base area (**i**); irregular vasculature (**j**) is also resolved in the lesion edge area, while the vasculature networks become more regular in the surrounding skin tissue (**k**). All images are color-coded to represent the two reconstructed frequency bands (red: larger structures in the bandwidth of 10–40 MHz; green: smaller structures in the bandwidth of 40–120 MHz). The three scans from the same patient with melanoma was measured once. All scale bar: 500 μm.

respectively. The dense pigmentation boundaries of the nevus and melanoma lesions appear in the coronal view MIP images of the EP layer (Fig. 4d, i, dashed lines). We observed dense and irregular vasculature, typical of melanoma[17,20], in the dermis layer of the tissue immediately surrounding the melanoma (Fig. 4j). In comparison, the vasculature in the tissue at the edge of the nevus appears comparable to healthy skin (Fig. 4e). We selected the edge areas of the lesions as regions-of-interest to compare the microvasculature between the nevi and melanomas (Fig. 4e, j, white dashed lines) because the reduced pigmentation minimizes attenuation due to melanin at the wavelength of 532 nm.

As a next step, we quantified vasculature features of the dysplastic nevi and melanomas to identify biomarkers for melanoma detection. The close surrounding tissue vessels (STV in Fig. 4e, j) in the dermis layers were segmented in a 500 μm distance extending out from the boundary of the pigmented lesion toward the healthy skin tissue. We then computed six biomarkers from these segmented vessels: (1) the total blood volume, (2) the vessel density, (3) the average vessel length, (4) the tortuosity, (5) the fractal number, and (6) the lacunarity (see Methods). The total blood volume and the vessel density metrics enable direct quantification of the amount of tumor-associated vascularity. The average vessel length and tortuosity indicated changes in the vascular geometry and morphology. The fractal number represents vascular spatial complexity associated with vascular changes and aberrant angiogenesis. Finally, the lacunarity characterizes tissue heterogeneity.

Figure 4k–p shows the results of the quantification and statistical comparisons of the six biomarkers. We observed significant differences of the six biomarkers characterized by two-tailed Mann–Whitney U-tests. The dermal vasculature of the melanoma edge areas exhibited a significantly higher total blood volume and vessel density compared with the melanocytic nevus, showing much higher vascularity of melanoma compared to nevi. The average total blood volumes for the melanoma and nevi were

$35.94\% \pm 6.22\%$ vs. $23.62\% \pm 5.61\%$, respectively (Fig. 4k, $P < 0.0001$) while the mean vessel densities were $0.017 \pm 0.004$ (a.u.) vs. $0.01 \pm 0.0035$ arbitrary units (a.u.) (Fig. 4l, $P < 0.0001$). The average length of the segmented vessels was $260.39 \pm 62.49$ μm in the nevus group and $139.60 \pm 46.63$ μm in the melanoma group (Fig. 4m, $P < 0.0001$), reflecting different spatial geometries of vessel patterns. The branched tree pattern of vasculature network was evident at the edge of the dysplastic nevi (Fig. 4e), while the vessel structure of the tissue surrounding the melanomas (Fig. 4j) was more tortuous and disorganized. These observations were confirmed by the tortuosity values of $0.27 \pm 0.082$ vs. $0.47 \pm 0.083$ (a.u.) for the nevus and melanoma groups, respectively (Fig. 4n, $P < 0.0001$). The melanoma lesions exhibited a higher mean fractal number of $1.26 \pm 0.18$ a.u., compared with $1.12 \pm 0.062$ a.u. for nevus lesions (Fig. 4o, $P = 0.02$), measuring vascular spatial complexity associated with vascular changes and aberrant angiogenesis. The melanoma lesions exhibited a higher mean lacunarity value of $0.170 \pm 0.083$ a.u., compared with $0.088 \pm 0.050$ a.u. for nevus lesions (Fig. 3p, $P = 0.0019$), indicating that the vasculature of melanoma is more inhomogeneous. We also found significant differences in the vasculature in the center of the lesions between the nevi and melanomas groups (Supplementary Fig. 4). In addition, we examined whether the combination of markers was useful in its ability to differentiate melanoma from nevus. Receiver operating characteristic (ROC) curves (Fig. 4q) were constructed based on the biomarkers of the TBV and tortuosity, revealing an area under the ROC curve (AUC) of 0.87 and 0.89 respectively, while the combinations of these two biomarkers achieved an area under the ROC curve of 0.93.

## Discussion
Skin microvasculature may have prognostic value in melanoma detection or determining its malignancy[18,45], however, conventional

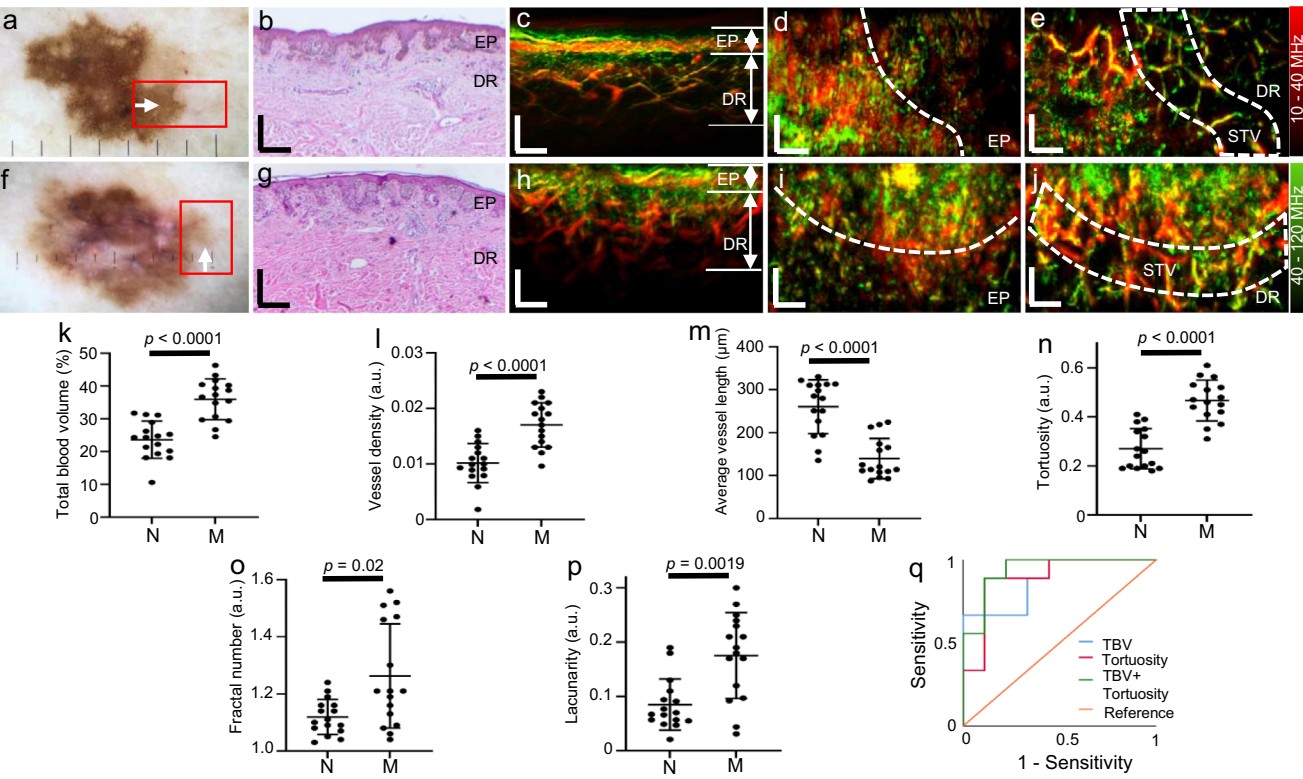

**Fig. 4 Vasculature feature quantifications between nevi and melanomas. a** Photograph of a dysplastic nevus from a patient back (male); the red rectangle indicates the scanning area. **b** Histological image of the dysplastic nevus corresponding to the area marked by the red rectangle in (**a**) and taken along the direction of the white arrow. **c** Cross-sectional MIP image measured at the edge area of the nevus marked by the red rectangle in (**a**). **d**, **e** Corresponding MIP images in the coronal direction of the epidermis (EP) and dermis (DR) layers of (**c**). **f** Photograph of a melanoma from a patient back (female); **g** Histological image of the melanoma corresponding to the area marked by the red rectangle in (**f**) and taken along the direction of the white arrow. **h** Cross-sectional MIP image measured at the edge area of the melanoma marked by the red rectangle in (**f**). **i**, **j** Corresponding MIP images in the coronal direction of the EP and DR layers of (**h**); White dash lines in (**d**, **i**) indicate the separation boundaries between the pigmented lesion and surrounding skin tissue in the nevus and melanoma. White dashed lines in (**e**, **j**) mark the vessel structure of the surrounding skin tissue, which is segmented by extending 500 μm length from the separation dash lines toward the healthy skin. STV: surrounding skin tissue vessels. All images are color-coded to represent the two reconstructed frequency bands (red: larger structures in the bandwidth of 10–40 MHz; green: smaller structures in the bandwidth of 40–120 MHz). **k–p**, the computed vessel biomarkers: total blood volume (**k**, 35.94 ± 6.22% vs. 23.62 ± 5.61%, $p < 0.0001$), the vessel density (**l**, 0.017 ± 0.004 vs. 0.01 ± 0.0035, $p < 0.0001$), average vessel length (**m**, 260.39 ± 62.49 vs. 139.60 ± 46.63, $p < 0.0001$), tortuosity (**n**, 0.27 ± 0.082 vs. 0.47 ± 0.083, $p < 0.0001$), fractal number (**o**, 1.12 ± 0.062 vs. 1.26 ± 0.18, $p = 0.02$) and lacunarity (**p**, 0.088 ± 0.050 vs. 0.170 ± 0.083, $p = 0.0019$) between the non-malignant nevi group (N, $n = 16$) and melanoma group (M, $n = 16$). **q** ROC plots in the differentiation of melanoma using TBV, tortuosity, and the combination (TBV + tortuosity). Data are expressed as the mean ± SD. Statistical significance was calculated using two-tailed Mann–Whitney U-tests. All patients were measured once. Source data are provided as a Source Data file. All scale bars: 500 μm.

imaging techniques lack the capabilities to fully exploit the microvasculature of melanoma lesions for these purposes. RSOM resolves human skin microvasculature morphology and quantifies biomarkers of skin diseases noninvasively and in vivo at greater depths and higher spatial resolutions than other optical methods. In this work, we constructed a fast RSOM (FRSOM) system capable of four times faster scanning than conventional RSOM, without a discernable loss of image quality or exceeding laser safety requirements, enabling single-breath-hold measurements and a significant reduction in motion artifacts. With the FRSOM, we for the first time comprehensively observed the microvascular morphology of melanoma on the human upper torso. Furthermore, our results demonstrate the ability of FRSOM to quantify the vascularity features in the adjacent tissue of melanoma, which had significantly increased density, complexity, and tortuosity compared to benign nevi.

Our previous implementation of RSOM scanned an area of $4 \times 2$ mm$^2$ in ~70 s because safety limits capped the maximum laser repetition rate at 500 Hz for a wavelength of 532 nm. However, FRSOM employs a high-sensitivity ultrasound detector

and a top coaxial illumination scheme to reduce the pulse energy of the illumination source by a factor of 4, affording a concomitant decrease in scan time to 15 s without exceeding laser safety limits. Therefore, FRSOM scans in a short enough time window for the patient to hold his or her breath, allowing the back and chest areas, which are severely impacted by breathing motion, to be imaged. As shown in Fig. 1 and Supplementary Fig. S1, the FRSOM demonstrated identical image quality and resolved similar skin morphology compared to the conventional RSOM. Moreover, single-hold-breath FRSOM (Fig. 2) enabled quantification of vasculature features of healthy and psoriatic skin at back and chest areas with strong motions beyond the capability of conventional RSOM with standard motion correction algorithm[40]. Several studies reported irregularly distributed vessel patterns of melanoma measured by dynamic optical coherence tomography. However, the reliable evaluation of the vascular morphology in melanomas based on OCT is limited to the superficial dermal layer at a depth of up to 500 μm[17,46]. FRSOM for the first time enables the microvasculature visualization of human melanoma and adjacent skin tissue on the upper torso at

unprecedented resolution and depth up to 1.5 mm at the wavelength of 532 nm. The significantly irregular and disordered vasculature architecture of melanoma in the center and at the boundary of the pigmented area is clearly resolved compared to the surrounding skin tissue as shown in Fig. 3 and Supplementary Movies S2–S4.

We further evaluated the feasibility of FRSOM to detect melanoma by quantifying and comparing the vasculature biomarkers from ten melanomas and ten nevi. FRSOM images clearly revealed the vasculature at the edge of the melanoma lesions to have a more irregular distribution compared to the nevi (Fig. 4). We selected tissue towards the edge of the lesion to quantify the vascular differences between melanomas and dysplastic nevi for two reasons: (1) to minimize the effects of the melanin signals at the lesion's center and (2) because previous studies have reported that angiogenesis occurs around tumors, altering the vasculature of the tissue bordering a melanoma lesion[22,31,47–49]. The edge areas of the scanned melanomas include parts of the tumors' tissue (tumor periphery as confirmed by histology), which exhibits significantly different vascular patterns compared to the adjacent healthy skin, as clearly illustrated in Fig. 3. In addition, dermoscopy (Fig. 3 and Fig. 4) indicated that the pigmentation is generally less dense at the edge of the lesions (tumor periphery) scanned in our study compared to the center of the lesions, affording FRSOM the highest possible penetration depth within the tumor area (about 1.5 mm at the edge areas as shown in Fig. 3). However, we also found significant differences between the vasculature in the centers of nevi and melanoma lesions, despite the higher light attenuation from melanin (Supplementary Fig. 4). Histological studies have revealed increased vascularity in melanomas when compared with nevi[18,20,22]. In agreement with these findings, two biomarkers (total blood volume and vessel density, see Fig. 4 and Supplementary Fig. 4) computed from the in vivo FRSOM images showed significantly higher values in the edges or in the center of the melanomas compared to the same areas in the nevi. The increased vascularity in the melanoma group also runs counter to expected reductions in skin microvasculature due to aging and photoaging[50], since our melanoma group had a significantly higher mean age compared to the nevi group, suggesting that these factors do not appreciably affect our results. In addition, previous OCT studies showed that melanomas display a more chaotic architectural organization of the superficial dermis layer in comparison to benign nevi or surrounding healthy skin[17,26,51]. Similarly, four FRSOM biomarkers (the average vessel length, tortuosity, fractal number, and lacunarity) calculated from the deeper dermal vasculature quantified obvious morphological differences between the melanoma and benign nevus groups. The comparisons of the six FRSOM biomarkers provides an initial set of vasculature features that are characteristic of melanoma and benign nevi, corresponding well to descriptions from conventional histopathology and OCT studies[17,26,51]. Therefore, FRSOM enables rapid, in vivo, noninvasive visualization of the microvascular architecture of melanoma lesions and adjacent skin tissue, which could increase the accuracy of melanoma detection and minimize the need for invasive biopsies. Furthermore, the increasing use of antiangiogenic drugs will stimulate demand for reliable, reproducible, and standardized means of assessing tumor angiogenesis response to these therapies[21]. FRSOM may provide a noninvasive alternative to repeated biopsies to assess the tumor vascularization of melanoma and monitor the response to antiangiogenic therapies[52].

The performance of multispectra RSOM is limited by a lack of portable ultra-fast pulsed lasers with pulse widths of 1–3 ns at sufficient energy. Spectral RSOM imaging has been demonstrated in healthy skin, but at a lower resolution and speed due to

employment of slow parametric oscillator lasers with pulse widths > 10 ns. For example, a scan of $4 \times 2$ mm$^2$ by our previous spectral RSOM implementation required ~13 min to record four wavelengths. In this work, the high-sensitivity transducer with preamplifier and coaxial fiber illumination allows implementation of spectral FRSOM by using four-wavelength laser source (1.4 kHz) with low pulse energy (10–25 μJ), which is significantly faster than our previous implementation using slow parametric oscillator laser at a repetition rate of 100 Hz[38]. We show that a 60 s scan with FRSOM resolves the distributions of melanin, deoxyhemoglobin, and oxyhemoglobin in healthy skin (Fig. 1d–g). Because this time frame is still too long for the measurement with single-breath-hold, we implemented dual-wavelength (515 nm and 532 nm) FRSOM to discriminate the melanin and hemoglobin distributions of healthy skin, nevi, and melanomas (Supplementary Fig. S2). Resolving melanin and hemoglobin can aid in identifying the boundary of a lesion to improve the selection of excision margins (Supplementary Fig. S2). The dual-wavelength FRSOM took about 25 s to scan the area of $4 \times 2$ mm$^2$, which enables single-breathing-hold spectra FRSOM measurements. Progress in ultra-sensitive ultrasound transducer and fast laser technology are essential to further decrease the acquisition time for single-breath-hold spectral FRSOM.

The results of our study are limited by the small number of subjects. Benign nevi in particular have very diverse features and a larger population is necessary to validate diagnostic performance. A more comprehensive study of a larger number of patients is necessary to validate the computed biomarkers and evaluate how well FRSOM features can distinguish melanoma from benign nevi. In addition, we investigated the vascular features of the close surrounding skin tissue of the melanocytic lesions, as the penetration depth of FRSOM is limited to about 1.5 mm with a laser wavelength of 532 nm, which may decrease when imaging the center areas of the pigmented lesion due to strong melanin absorption (Fig. 3). We found that FRSOM can resolved the vasculature at depths of >1.2 mm at the center of a pigmented lesion and >1.5 mm in the surrounding skin (see Fig. 3 and the Supplementary Movies S2–S4). When compared to the results of the vascular biomarkers computed from images acquired from the edge areas of the lesions (Fig. 4), these differences in the lesion center between the nevi and melanoma group (Supplementary Fig. 4) are less significant (for example, for the biomarker total blood volume $p = 0.001$ vs. $p = 0.01$). This discrepancy may result from the segmented dermal vessels in the center pigmented areas being more affected by melanin signals than at the edges. In order to better visualize and quantify the vasculature under the pigmented areas, multispectra or dual-wavelength FRSOM can be used to separate melanin signals from the vascular signals as shown (see Fig. 1 and Supplementary Fig. 2), which may allow direct computation of vascular biomarkers without segmenting the boundaries. In the future, we plan to improve FRSOM's clinical capabilities using the multi-spectral approach.

SNR variations and motion artifacts can significantly affect the determination of the lesion boundary and vessel segmentation, which may degrade the accuracy of the vascular biomarker computation. For all FRSOM measurements, the data quality and SNR of the measurements were pre-screened via a quality control scheme and only high-quality datasets were employed in subsequent analyses (see Data quality control in Methods). In the future, the signal quality of recorded FRSOM data can be further evaluated by taking account the skin tone or ulceration of a lesion[53] while the motion should be monitored during the whole scanning period to avoid random movements of patients[40]. The accuracy of vascular biomarker computation is determined by the

boundary and vessel segmentation method. Although our segmentation methods and biomarker computation achieve robust results (see Validation of the boundary and vessel segmentation methods in Methods), advanced data processing methods, like deep learning-based approaches[54], can be applied to directly interpret FRSOM images for diagnosis in the future.

In conclusion, we built a single-breath-hold FRSOM imaging system, which enables noninvasive visualization of the micro-vasculature of pigmented melanocytic lesions. The quantitative assessment of the vascular biomarkers exhibited significant differences of the vascular morphology between the nevi and melanoma groups. Although in vivo analysis and larger-scale studies are required to further validate the technique's capability and feasibility, we believe that the results of our initial pilot study show great potential for FRSOM imaging as a preoperative screening tool, providing complementary information for melanoma detection as well as in the longitudinal monitoring of treatment options.

## Methods

**The FRSOM system**. The FRSOM system (Supplementary Fig. 1a) was implemented based on a custom-made spherically focused LiNbO₃ through-hole transducer and a coaxial illumination scheme achieved by inserting a single multimode fiber (Thorlabs, 200 μm in diameter) through the central aperture (350 μm in diameter) of the transducer. In our previous implementation of RSOM, the bundle illumination scheme (Supplementary Fig. 1a) achieved a homogeneous illumination pattern with diameter of 2.5 mm and restricted the repetition rate of laser source at the wavelength of 532 nm under 500 Hz because of the human-use safety limits described in the American National Standard for Safe Use of Lasers (ANSI Z136.1-2014, Laser Institute of America)[31]. However, the coaxial illumination through single fiber of FRSOM obtained a circle illumination pattern of 1.4 mm in diameter (Supplementary Fig. 1a) and enabled to reduce the pulse energy by a factor of four at a value of 18 uJ and correspondingly increased the laser repetition rate of 1.4 kHz without dropping the light fluence within the illuminated area. In addition, the novel through-hole transducer transducer was designed and implemented by directly mounting an integrated miniature preamplifier (30 dB amplification, ERA-8SM+, Mini-Circuits, Brooklyn, New York) on the ultrasound transducer before the signal was transmitted to the acquisition card (Gage, CSE161G2, 1 GS/s, US). In addition, we built an impedance-matched transmission of the amplified signal that was less affected by the long transmission distance, which gained two times of SNR by minimizing artifacts compared with the transducer without preamplifier used for the conventional RSOM system (Supplementary Fig. 1b). Our previous implementation of RSOM scanned over 4 mm × 2 mm (scan points, 266 × 135), taking 70 s. However, FRSOM images were reconstructed using optoacoustic signals collected over 4 mm × 2 mm (scan points: 201 × 101) with a total scanning time of 15 s, achieving about four times scanning speed acceleration. The technical comparisons between conventional RSOM and FRSOM are listed in supplementary Table 1. The imaging performance comparison between RSOM and FRSOM was characterized by measuring the same skin area at the forearm of a healthy volunteer sequentially, as shown in Supplementary Fig. 1c–h. We observed that the skin morphology (epidermis and dermis layers) and dermal vasculature were clearly resolved by both implementations with identical image quality. For the FRSOM vascular imaging, we use the single wavelength of a 532 nm laser (Bright Solutions, Italy) with temporal width of the pulse 0.9 ns and maximum repetition rate of 2 kHz. Besides, we further implemented multispectral FRSOM by using a custom-designed four wavelengths (532, 555, 579, and 606 nm) laser source with maximum repetition rate of 1.4 kHz and low pulse energy (10–25 μJ) at the fiber output. The multispectral FRSOM took about a total of 60 s to scan over 4 mm × 2 mm (scan points for each wavelength: 201 × 101), which exceeded a single-breath-hold. Further, dual-wavelength (515 nm and 532 nm) FRSOM was implemented to further reduce the scanning time of about 25 s for the area of 4 × 2 mm² (scan points for each wavelength: 201 × 101). We have demonstrated that the dual-wavelength FRSOM can discriminate the melanin and hemoglobin distributions of healthy skin, nevus, and melanoma as shown in the Supplementary Fig. S2. The dual-wavelength FRSOM could enable single-breathing-hold spectra FRSOM measurements for some individuals who could hold breath for 25 s.

**Motion correction and image reconstruction**. Our previously developed motion correction method has been applied to correct motions of the recorded FRSOM data[40]. In Fig. 2a–d, the motion graphs between the normal breathing and hold-breathing were computed by transforming the two-dimensional motion matrix into one dimension. At the illumination wavelength of 532 nm, the amplitude of the one-dimensional (1D) signal was high when the transducer sensed the part of the wavefront generated at the melanin layer in the stratum basale and the micro-vasculature in the different areas of the dermis. Therefore, the motion graphs

captured the vertical displacements of the stratum basale and microvasculature. By computing the position of the maximum of the cross-correlation function between the first 1D signal and the remaining 1D signals, one can calculate a displacement function. The detail information about the motion computation has been introduced in our previous work[40]. For the image reconstruction, the motion-corrected FRSOM signals were divided into two frequency bands 10–40 MHz (low) and 40–120 MHz (high) for the 10–120 MHz bandwidth. Signals in the two different bands were independently reconstructed. The beam-forming method was used to reconstruct three-dimensional images[55]. The reconstruction algorithm was accelerated by parallel computing on a graphics processing unit and improved by incorporating the spatial sensitivity field of the detector as a weighting matrix. The reconstruction time of one bandwidth took about 2 min with voxel size of the reconstruction grid at 12 μm × 12 μm × 3 μm. The two reconstructed images $R_{low}$ and $R_{high}$ corresponded to the low- and high-frequency bands. A composite image was constructed by fusing $R_{low}$ into the red channel and $R_{high}$ into the green channel of a same RGB image. The detail process has been introduced in our previous work[30]. The FRSOM images can be rendered by taking the maximum intensity projections (MIPs) of the reconstructed images along the slow axis or the depth direction as shown in Figs. 1 and 2.

**Study population, histology, and general statistics**. We measured two healthy volunteers and a patient with psoriasis. In addition, twenty lesions (ten melanomas from ten patients and ten nevi from nine patients) were imaged. All human measurements were followed approval from the Ethics Committee of Klinikum Rechts der Isar der Technischen Universität München, Munich, Germany. The melanoma patient group (six females and four male) had an average age of 70.0 ± 14.6 years while the nevus group (four females and six male) had an average of 48.5 ± 18.6 years. The measured lesions were located on the back, chest, upper shoulder, arm, and leg. The tumor thicknesses ranged from 0.2–1.8 mm while the thicknesses of the nevi ranged from 0.23–1.3 mm. The detailed characteristics of all patients are listed in supplementary Table 2. According to the Breslow Index in Melanoma, the thickness of the nevi was calculated from the directly overlaying stratum granulosum to the deepest point of the nevus, e.g., to the basal point of a nest. All participants gave written informed consent before the measurement. During the FRSOM scanning, patients were asked to hold breath for 15 s in order to reduce motions. Low quality datasets due to serious motion artefacts were excluded from the study.

All lesions were diagnosed by professional dermatologists and pathologists based on the histology image. CD31 immunostaining was also performed to evaluate vessel footprints. All healthy volunteers and patients were measured independently by the same imaging system and the reproducibility of the imaging system has been fully validated[30]. To assess the significance of the statistical differences for the metrics used to compare melanoma and nevus, we performed two-tailed Mann–Whitney U-tests. All statistical analyses were performed with GraphPad Prism 9.0 software (GraphPad Software Inc., San Diego, California, USA), with P-values < 0.05 considered statistically significant. The selection of the nevus and melanoma areas to be imaged was made by professional dermatologists, independently of the authors that processed the data. The diagnostic accuracy of independent marker or the combination of two markers was assessed with multivariate logistic regression and analyzed using ROC curves and by calculating the AUC.

**Data quality control**. Our previous studies have shown that motion can significantly affect image quality, although our motion correction algorithms can significantly mitigate this effect[40,41]. However, various motions from physiological displacements due to arterial pulsation and heartbeat and unintentional movements of the patient may lead to inconsistent motion correction improvements. Even though FRSOM enables 15 s single-breath scans that can significantly minimize overall motion, random movements of patients may still occur during the scanning period. Therefore, we applied a quality control scheme based on the amount of motion in the raw data that classifies the quality of data collected. The quality control scheme enables the selection of high-quality datasets, in which the motion is minimal enough for the motion correction algorithm to correct, resulting in more uniform image quality for quantitative analysis[41].

**Lesion boundary and vessel segmentation and biomarker calculation**. With the 3D reconstructed FRSOM image, the skin surface was first flattened (Supplementary Fig. 3a). The corresponding MIP images in the coronal direction of the epidermis (EP) resolved the dense pigmentation distribution of the lesion. The boundaries between lesions and adjacent skin tissue were determined by the contrast differences between the pigmented lesion and the surrounding tissue, as the pigmented area generates much stronger optoacoustic signals compared to the surrounding skin. The dense and high contrast pigmented structures of the epidermis layer in the MIP image were automatically segmented by a graph theory and dynamic programming-based approach[56] (Supplementary Fig. 3b, dashed lines). Based on the segmented boundaries, we further separated the dermis vasculature (typically a volume of 4 mm × 2 mm × 2 mm was selected, although the dermis depth varied by patient) between the tumor base and the surrounding skin tissue (Supplementary Fig. 3c). The surrounding tissue vessels (STV) in the dermis

layer was defined as 0.5 mm length extension from the boundary dash lines toward the healthy skin as shown in Supplementary Fig. 3d. To quantify the features of STV, we further segmented the vessel boundaries (Supplementary Fig. 3e), the skeleton and branching points (Supplementary Fig. 3f) based on the AngioQuant method[57].

The total blood volume (TBV) is determined as the ratio $TBV = N/V$, where $N$ is the number of nonzero voxels from the STV image volume and $V$ is the total segmented volume. The vessel density is calculated from the two-dimensional segmented STV image (Supplementary Fig. 2e) as the ratio between the total nonzero pixels of STV and the whole segmented area. The average vessel length is calculated as the ratio between the sum lengths of all vessel skeletons and the total number of branching points (indicated as the vessel number). The tortuosity of the curved vasculature is computed based on the distance metric (DM) defined as the ratio between the actual path length (AL) of a vessel and the straight-line distance (SL) between two end points of this blood vessel: $DM = \frac{AL}{SL}$[58]. The fractal dimension number (FN) describes the complexity of an irregular object[59]: $FN = -lim_{r \to 0} \frac{\log[N(r)]}{\log(r)}$, where $N(r)$ represents the number of cubes required to cover an object when the cube side length is $r$. Generally, the higher the complexity of an object, the higher the fractal dimension number. In addition, the lacunarity (L) can be used to characterize the "lumpiness" of the fractal data, providing metainformation about the computed FN values[59], defined as: $L = \frac{1/MN \sum_{m=0}^{M-1} \sum_{n=0}^{N-1} I(m,n)^2}{\{1/MN \sum_{k=0}^{M-1} \sum_{l=0}^{N-1} I(k,l)\}^2} - 1$, where $M$ and $N$ are the size of the FN processed image $I$. The higher the lacunarity, the more inhomogeneous the examined fractal areas and vice versa[59].

**Validation of the boundary and vessel segmentation methods**. In order to test the boundary segmentation method described above, we validated the results against manual segmentation performed by two well-trained observers independently. The correlation coefficients between the auto and manual segmentations were 0.91 (Observer 1) and 0.93 (Observer 2). In addition, our previous work has done comparison of capillary imaging by conventional nailfold capillaroscopy and RSOM[36]. The capillary diameter and capillary density computed from the segmented RSOM images correlated well with the nailfold capillaroscopy. In addition, our approaches to segment RSOM skin features (including the epidermis layer thickness and dermal vasculature) were validated in previous work by histology[30].

**Reporting summary**. Further information on research design is available in the Nature Research Reporting Summary linked to this article.

## Data availability

The raw optoacoustic imaging data of patients with melanoma and nevi that support the findings of this study are available from the corresponding author upon request, with permission of the Klinik und Poliklinik für Dermatologie und Allergologie am Biederstein, Munich, Germany. The raw optoacoustic data for characterizing the transducers and motion effects is shared in the public repository: https://doi.org/10.5281/zenodo.6466446. Source data are provided with this paper.

## Code availability

The image reconstruction and processing algorithms are described in detail in Methods and in the reference. The full reconstruction code is not publicly available because the code is used in licensed technology. The main function of the image reconstruction code is shared in this public repository: https://doi.org/10.5281/zenodo.6466446.

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

## Acknowledgements
This project has received funding from the European Union's Horizon 2020 research and innovation program under grant agreement No 687866 (INNODERM) and under grant agreement No 871763 (WINTHER). We thank Dr. Robert J. Wilson for his advice and editing and the staff at Department of Dermatology and Allergy at TUM for assisting with the patient studies.

## Author contributions
H.H. developed the imaging system, designed and performed the experiments, processed the data, provided conceptual input, and wrote the paper. C.S. recruited the patients, performed the patient experiments and provided conceptual input. M.S. developed the imaging system. B.H. recruited the patients and helped with the patient experiments. A.B. helped to perform the experiments. A.S.-G. performed the histology experiments. U.D. provided conceptual input. J.A. provided conceptual input, developed the imaging system, designed the experiments, contributed to writing the paper, and led the research. V.N. provided conceptual input, provided funding and wrote the paper.

## Funding

## Competing interests
V. N. is an equity owner in and consultant for iThera Medical GmbH, Munich, Germany. iThera Medical GmbH helped to accelerate the image reconstruction code that the authors developed. All other authors declare no competing interest.
