## [Peer Review File · Nature Communications]

Reviewers' comments:

Reviewer #1 (Remarks to the Author): Expert in optoacoustic imaging

General comments

In this manuscript, the authors present their recent iteration of raster scan optoacoustic mesoscopy with crucial hardware and software upgrades that enable them to acquire single-breath-hold scans. With this technology development, they apply the imaging to a cohort of patients, and demonstrate that there are large differences between malignant and benign skin cancer lesions. As such, this is a significant advance. The approach builds upon the previous publications that the authors presented to the community (29).

The study is preliminary in its scientific findings Due to the limited number of subjects and diverse phenomenology of benign nevi, as the authors attest. However, the significance lies in that this imaging modality is the only one that can achieve the spatial resolution and imaging depth to image the lesions fully and to provide vascular morphometric measurements of this level of sophistication. OCT might have a better resolution; however, the modality cannot image at depth and differentiate between the melanin and the vascular structures.

The insights are interesting and warrant publication. However, several technical details need to be clarified, the data needs to be clarified, and a more in-depth discussion of the findings is warranted, especially compared to the findings of other imaging approaches with a comparison of other approaches (e.g. polarization-sensitive OCT etc.).

Specific comments

1. A significant part of the innovation is technical, and comparisons are made between the two - perhaps a table comparing FRMOM and RMOM should be added in the appendix to understand the differences in the two approaches better, focusing on key advantages (e.g. what is SNR gain?).
2. Line 118: In figure 1 b,c: has the colour coding been explained at this point in time? What does it represent? For e-g, what do the different colors represent quantitatively, and what is purple? If the images in the right panel are acquired at different locations from the images in the panel on the left, that should be explicitly stated (b,c) – while if d is at the same location as e and f with g, again, this should be explicitly stated. Finally, while capillaries have been detected in the cuticle, it is not for the "skin structures" I assume. Where were the skin structures imaged? This should also be commented on. Finally, for f and g, shouldn't there also be a vertical scale bar, as in the figure above?
3. Line 150 / Figure 2: How was the maximum of 300 μm determined? It seems larger than that to me upon examination of 2c. Moreover, there seems to be high-frequency periodicity in 2b/d. What causes this motion (that is successfully corrected)? Finally, green here does not present melanin but "dilated capillary loops". Is this the case? How was the colour-coding done in this example?
4. Figure 3, Caption: What is the orientation of the H&E stain compared to the image in a? I cannot see white dashed lines in (d). Do you mean g,j? Maybe you can have the white dashed lines also superimposed on image (a). What is green in this image (it should be in the caption)? For the EP layer in

g, why is there no signal represented in the image outside the white dotted line? Scan 3 (normal) in (h) would indicate that there should be a signal. Is the dynamic range kept constant? C-e: Which cross-sections are imaged within the red rectangle? Can dotted lines be introduced in the rectangle if the location is known?

5. Figure 4, Caption: Same question re: H&E stain location as in Fig.3. Based on what criterion is the location/shape of the white line determined? The aggregate data in k-q: what was the ROI from which the data were collected for the N and M groups? What does each dot represent? Finally, why was the STV in the DR chosen as the location to perform the analysis (understood only after reading the main text)? Where other locations considered in trying to identify the imaging biomarkers? To generate the ROC plots, why were TBV and tortuosity chosen?

6. What was the age distribution/sex of the patients for which the melanomas and nevi were collected?

7. Line 412: Is Fig 1a meant instead of b? Or Supplementary 1b?

8. L442: Why were these four wavelengths chosen?

9. L487: while the method for determining the lesion boundary and performing the vessel segmentation is highlighted, more discussion is warranted about how particular choices are made that impact boundary delineation and the data collected. Some insight is warranted, as these are critical to the development of vascular biomarkers. How sensitive are these to the SNR of the measurements? To the region/volume of interest? To the motion? Do variations in choices impact the conclusions? As the technology matures and new communications arise, emphasizing the rigour of the data analysis, the data analysis pipeline, and the imaging biomarker's robustness becomes more critical. This includes linking the biomarker features to the biophysical interpretation based on histology and what is known of key absorbing structures.

Reviewer #2 (Remarks to the Author): Expert in melanoma imaging

For the first time, the study uses a fast optoacoustic method (FRSOM) to examine the blood vessels in malignant melanomas in vivo. A small number of 10 melanomas are compared with 10 nevi. The measurements are taken while the patient is holding their breath to minimize motion artifacts. The blood vessels in the skin just outside of the lesion are measured because the melanin in the center covers the vascular signal. Parameters such as vessel density, shape and branching are calculated on the three-dimensional images. The authors found significant differences in the vascular patterns between melanomas and nevi.

1. No information is given on the histological parameters of the lesions. Melanomas are extremely heterogeneous. There are different subtypes. The tumor thickness is an essential parameter for the vessel density and morphology, as could be shown in studies on dynamic OCT of melanomas. Nevi are very heterogeneous, too, also in terms of thickness, pigmentation and subtype. It remains unclear whether all of the nevi were dysplastic. Since the thickness and pigmentation of a lesion certainly have a major influence on the representation of the vessels in the FRSOM, the differences found can also only be based on the fact that the melanomas were thicker or more pigmented than the nevi.

2. It was not measured in the center of the lesion, but directly outside the melanoma or the nevus in healthy skin: „We selected the vessels in the skin tissue just outside of the lesions (Fig. 4e,j, white dash

lines) as regions-of-interest to compare the microvasculature between the nevus and melanoma, since the vasculature optoacoustic signals under the pigmented area may be attenuated by the melanin signals at the wavelength of 532 nm.”

In the center, imaging with vascular signal using RSOM is apparently not possible because the melanin absorbs too much signal. Since the vascular polymorphism correlates with the thickness, it can be assumed that the thick center of a lesion has more conspicuous vessels than the flat macular margin or the healthy skin at the border. The clinical value of the method is therefore very questionable if it cannot be measured at the most pronounced part of the melanoma. Then the higher penetration depth of RSOM compared to dynamic OCT has no added value. The studies in which D-OCT was used to visualize melanoma vessels compared images from the center of the lesion, where the tumor vessels could be visualized despite the lower penetration depth of the OCT signal. Apparently, the vessels within a pigmented lesion cannot be visualized using RSOM, only perilesional vessels. It is not clear why the informative value of this technology should be greater than that of dynamic OCT, which also has a higher resolution. In addition, dynamic OCT has proven itself in everyday clinical practice. An in vivo measurement is possible without any problems on the back or trunk without holding breath, because the measurement is very quick and the handpiece is so light and flexible that it can follow the movements. The only advantage of RSOM over D-OCT, the greater depth of penetration, apparently cannot be used in pigmented lesions.

A formal point of criticism is that the structure of the chapters does not correspond to the content. Results are already presented in the introduction; the methodology is explained in the results section. The chapters need to be sorted better.

Significant recent work on blood vessel imaging in melanoma was not cited:

Welzel J, Schuh S, De Carvalho N, Themstrup L, Ulrich M, Jemec GBE, Holmes J, Pellacani G. Dynamic optical coherence tomography shows characteristic alterations of blood vessels in malignant melanoma. *J Eur Acad Dermatol Venereol*. 2021 May;35(5):1087-1093. doi: 10.1111/jdv.17080. Epub 2021 Jan 5. PMID: 33300200.

Reviewers' comments:

Reviewer #1 (Remarks to the Author): Expert in optoacoustic imaging

General comments

In this manuscript, the authors present their recent iteration of raster scan optoacoustic mesoscopy with crucial hardware and software upgrades that enable them to acquire single-breath-hold scans. With this technology development, they apply the imaging to a cohort of patients, and demonstrate that there are large differences between malignant and benign skin cancer lesions. As such, this is a significant advance. The approach builds upon the previous publications that the authors presented to the community (29).

The study is preliminary in its scientific findings Due to the limited number of subjects and diverse phenomenology of benign nevi, as the authors attest. However, the significance lies in that this imaging modality is the only one that can achieve the spatial resolution and imaging depth to image the lesions fully and to provide vascular morphometric measurements of this level of sophistication. OCT might have a better resolution; however, the modality cannot image at depth and differentiate between the melanin and the vascular structures.

The insights are interesting and warrant publication. However, several technical details need to be clarified, the data needs to be clarified, and a more in-depth discussion of the findings is warranted, especially compared to the findings of other imaging approaches with a comparison of other approaches (e.g. polarization-sensitive OCT etc.).

Re: We thank the reviewer for the positive comments and suggestions. We have endeavored to address all of the technical comments listed below. **All changes in the text of the revised paper are marked in red.** In addition, we added more in-depth discussion in the context of other melanoma studies based on histological analysis and D-OCT:

“FRSOM images clearly revealed the vasculature at the edge of the melanoma lesions to have a more irregular distribution compared to the nevi (Fig. 4). We selected tissue towards the edge of the lesion to quantify the vascular differences between melanomas and dysplastic nevi for two reasons: 1) to minimize the effects of the melanin signals at the lesion’s center and 2) because previous studies have reported that angiogenesis occurs around tumors, altering the vasculature of the tissue bordering a melanoma lesion¹⁻⁵. The edge areas of the scanned melanomas include parts of the tumors’ tissue (tumor periphery as confirmed by histology), which exhibits significantly different vascular patterns compared to the adjacent healthy skin, as clearly illustrated in Fig. 3. In addition, dermoscopy (Fig. 3 and Fig. 4) indicated that the pigmentation is generally less dense at the edge of the lesions (tumor periphery) scanned in our study compared to the center of the lesions, affording FRSOM the highest possible penetration depth within the tumor area (about 1.5 mm at the edge areas as shown in Fig. 3). However, we also found significant differences between the vasculature in the centers of nevi and melanoma lesions, despite the higher light attenuation from melanin (supplementary Fig. 4). Histological studies have revealed increased vascularity in melanomas when compared with nevi^{1,6,7}. In agreement with these findings, two biomarkers (total blood volume and vessel density, see Fig. 4 and supplementary Fig. 4) computed from the in vivo FRSOM images showed significantly higher values in the tissue surrounding or in the center of the melanomas compared to the same areas in the

nevi. In addition, previous OCT studies showed that melanomas display a more chaotic architectural organization of the superficial dermis layer in comparison to benign nevi or surrounding healthy skin⁸⁻¹⁰. Similarly, four FRSOM biomarkers (the average vessel length, tortuosity, fractal number, and lacunarity) calculated for the deeper dermal vasculature quantified obvious morphological differences between the melanoma and benign nevus groups. The comparisons of the six FRSOM biomarkers provides an initial set of vasculature features that are characteristic of melanoma and benign nevi, corresponding well to descriptions from conventional histopathology and OCT studies⁸⁻¹⁰."

Specific comments

1. A significant part of the innovation is technical, and comparisons are made between the two - perhaps a table comparing FRSOM and RSOM should be added in the appendix to understand the differences in the two approaches better, focusing on key advantages (e.g. what is SNR gain?).

Re: Yes, we have adopted the reviewer's suggestion and have inserted a table (Supplementary Table I) comparing the technical differences between the FRSOM and conventional RSOM including the SNR, scanning speed, illumination laser energy, and illumination spot size. In the Methods section describing the FRSOM system and supplementary Fig. 1, we describe two times increase in SNR for the FRSOM system compared to the conventional RSOM system because of minimizing artifacts and the usage of preamplifier. In addition, the image quality between FRSOM and RSOM has been compared by measuring the same human skin area as shown in the supplementary figure 1, which demonstrates that the FRSOM achieves comparable image quality to that of the conventional RSOM.

2. Line 118: In figure 1 b,c: has the colour coding been explained at this point in time? What does it represent? For e-g, what do the different colors represent quantitatively, and what is purple? If the images in the right panel are acquired at different locations from the images in the panel on the left, that should be explicitly stated (b,c) – while if d is at the same location as e and f with g, again, this should be explicitly stated. Finally, while capillaries have been detected in the cuticle, it is not for the "skin structures" I assume. Where were the skin structures imaged? This should also be commented on. Finally, for f and g, shouldn't there also be a vertical scale bar, as in the figure above?

Re: We thank the reviewer for these detailed comments and have improved the text in the revised version (marked in red). We have added a color bar in the Fig .1b and 1c to explain the color coding. For the image reconstruction, FRSOM signals were divided into two frequency bands 10-40 MHz (low frequency) and 40-120 MHz (high frequency) for the 10-120 MHz bandwidth. Signals in the two different bands were independently reconstructed. In Fig. 1b and 1c, the color coding represents the low (10-40 MHz, large structures)- and high-frequency bands (40-120 MHz, fine structures). A composite image was constructed by fusing reconstruction of low frequency into the red channel and reconstruction of high frequency into the green channel of a same RGB image. The detailed process was introduced in our previous work (Aguirre et al. Nature Biomedical Engineering, 2017). This information has been also discussed in the Methods section. We agree with the reviewer that the color-coding should be better explained in the text referring to Fig. 1; the added information is marked in red.

We have added color bar to explain the color in Fig. 1d-g as well. In Fig. 1d-g, the reconstructed images were unmixed for melanin (green), oxyhemoglobin (red) and deoxyhemoglobin (blue), the purple

color is the overlay between oxyhemoglobin (red) and deoxyhemoglobin (blue) in the maximum intensity projection image. This information has been better described in the description of Fig. 1 and in the caption of Fig. 1. The images (b,c) are the maximum intensity projections of different views, which were acquired from the same location; Fig. 1b is the top view while Fig. 1c is the side view. The images (d-g) were also acquired from the same location and visualized with different views, as the same vasculature are clearly resolved in all images. In addition, by using multi-spectra FRSOM, the melanin (green), oxyhemoglobin (red), and deoxyhemoglobin (blue) are quantified and visualized. Right, we measured the capillaries (in the diameter of 10 – 20 μm) of the cuticle to demonstrate the superior performance of FRSOM imaging quality. Fig. 1d-f show the skin structures, as does supplementary figure 1. We have now added a vertical scale bar in f and g as well as other figures in the paper.

3. Line 150 / Figure 2: How was the maximum of 300 μm determined? It seems larger than that to me upon examination of 2c. Moreover, there seems to be high-frequency periodicity in 2b/d. What causes this motion (that is successfully corrected)? Finally, green here does not present melanin but "dilated capillary loops". Is this the case? How was the colour-coding done in this example?

Re: By calculating the position of the maximum of the cross-correlation function between the first 1D signal and the remaining 1D signals, we can compute a displacement function to extract the motions of the FRSOM data. The detail information about the motion graph computation has been introduced in our previous paper (Aguirre et al. IEEE Transactions on Medical Imaging, 2019). The maximum of 300 μm is determined by summing the absolute motion values in the positive and negative directions. The reviewer is correct that the maximum value should be around 350 μm . This oversight has now been corrected in the revised version. The method of the motion graph computation has been added to the results and methods sections. The high-frequency periodicity was generated by computing the cross-correlation between each 1D signal from the human skin with the certain curvature. The details have been also described in our previous work as well (Aguirre et al. IEEE Transactions on Medical Imaging, 2019). Yes, these motions have been smoothed and corrected. For FRSOM image reconstruction, the green color represents the high frequency detailed structures, like the capillary loops, while the red color represents larger structures, like the melanin layer and the large dermal vessels. We have added a color bar in the Fig. 2. Detailed information about the color coding reconstruction has been introduced in the method section and our previous paper (Aguirre et al., Nature Biomedical Engineering, 2017).

4. Figure 3, Caption: What is the orientation of the H&E stain compared to the image in a? I cannot see white dashed lines in (d). Do you mean g,j? Maybe you can have the white dashed lines also superimposed on image (a). What is green in this image (it should be in the caption)? For the EP layer in g, why is there no signal represented in the image outside the white dotted line? Scan 3 (normal) in (h) would indicate that there should be a signal. Is the dynamic range kept constant? C-e: Which cross-sections are imaged within the red rectangle? Can dotted lines be introduced in the rectangle if the location is known?

Re: the histology image was taken along the center line of the red rectangle in the direction indicated by the black arrow 1 in Fig. 3a, which correlates to the direction of the cross section image of Fig. 3c. The three images in Fig. 3 were acquired at the locations indicated by the three black arrows and the image field-of-view was marked by the red rectangle. In response to the reviewer's suggested, we have superimposed a white dashed line onto image (a), which indicates the boundary between the pigmented lesion and the surrounding tissue. The green color represents the smaller structures. We

have added a color bar in the figure. Yes, because of the defined dynamic range, the epidermis signal outside the white dotted line was much less compared to the signals within the pigmented area. The cross sectional images (c-e) are the maximum intensity projections along the short scanning distance of the red rectangle. The red rectangle indicates the scanning field-of-view of each location. The dotted lines represent the boundary of the pigmented lesion and the surrounding tissue, which were generated by the large differences in melanin signals between the pigmented lesion and the surrounding skin. The dotted lines can be introduced in the rectangle by marking the edge of the pigmented area as shown in image (Fig. 3a), and the similar boundaries of the pigmented area were resolved in the FRSOM images (Fig. 3g).

5. Figure 4, Caption: Same question re: H&E stain location as in Fig.3. Based on what criterion is the location/shape of the white line determined? The aggregate data in k-q: what was the ROI from which the data were collected for the N and M groups? What does each dot represent? Finally, why was the STV in the DR chosen as the location to perform the analysis (understood only after reading the main text)? Where other locations considered in trying to identify the imaging biomarkers? To generate the ROC plots, why were TBV and tortuosity chosen?

Re: the location of the H&E stain is now indicated by white arrows in Fig. 4. The white line is determined by the significant contrast differences between the pigmented lesion and the surroundings, as the pigmented lesion contains much denser melanin signals compared to the surrounding skin. The dense and high contrast pigmented structures of the epidermis layer in the MIP image was automatically segmented using a graph theory and dynamic programming based approach. This information is better described in the Methods section. The ROI for the surround tissue vessels (STV) extends 500 μm from the segmented white dotted line (edge of the lesion) towards the healthy skin. We computed the vasculature biomarkers in the ROI to quantify the vascular differences between the nevus and melanoma groups. Each dot represents the computed biomarker value from one FRSOM image, acquired either from patients with nevi or melanoma. We choose the STV values for two reasons: 1) To minimize the effects of the melanin signals at the lesion center; 2) Previous studies have reported that angiogenesis occurs around tumors and the vasculature of the adjacent tissue around the melanoma lesion could be altered¹⁻⁵. In addition, the edge tissue of the melanoma we scanned includes parts of the tumor tissue, which exhibits significantly different vascular patterns compared to the adjacent healthy skin as clearly illustrated in Fig. 3. This is actually one of the key findings of our paper. The selected imaging areas were suggested by three expert dermatologists and were afterwards confirmed by histology.

We have now compared and computed the vasculature biomarkers inside the pigmented lesion between the nevi and melanomas groups, as shown in the new supplementary figure 4, which indicates slightly less significant differences compared to results of Fig. 3. The reason for this difference could be that the segmented dermal vessels in the center pigmented areas could be more affected by melanin signals than at the edges. FRSOM can resolved the vasculature at the depth of >1.2 mm inside the pigmented lesion and >1.5 mm in the surrounding skin as shown in Fig. 3 and the supplementary videos. We also discussed the possible solutions to improve the results in the discussion: *“This discrepancy may result from the segmented dermal vessels in the center pigmented areas being more affected by melanin signals than at the edges. In order to better visualize and quantify the vasculature under the pigmented areas, multi-spectra or dual-wavelength FRSOM can be used to separate melanin signals from the vascular signals as shown (see Fig. 1 and supplementary Fig. 2), which may allow direct computation of vascular biomarkers without segmenting the boundaries. In the future, we plan to improve FRSOM’s clinical capabilities using the multispectral approach.”*

We choose TBV and tortuosity to compute the ROC curve, because the computation of total blood volume and the tortuosity are actually very simple and robust and they show significant differences between the two patient groups.

6. What was the age distribution/sex of the patients for which the melanomas and nevi were collected?

Re: the age distribution and sex of the patients have now been added to the Methods sections, as well as the subtype and thickness of the lesions as determined by histology. The detailed characteristics of all patients have been added in the supplementary Table II.

7. Line 412: Is Fig 1a meant instead of b? Or Supplementary 1b?

Re: We have now corrected this error. It should be Supplementary Fig. 1a.

8. L442: Why were these four wavelengths chosen?

Re: the four wavelengths were chosen based on the absorption spectrum of oxy- and deoxyhemoglobin, which are optimal points (significant absorption differences between oxy- and deoxyhemoglobin at wavelength 555 nm and 606 nm) to solve the unmixing problem in order to compute the concentration of melanin, oxy- and deoxyhemoglobin. In addition, the wavelength choices were influenced by the availability of a laser that was capable of producing both enough pulse laser energy and high repetition rate at the visible wavelengths.

9. L487: while the method for determining the lesion boundary and performing the vessel segmentation is highlighted, more discussion is warranted about how particular choices are made that impact boundary delineation and the data collected. Some insight is warranted, as these are critical to the development of vascular biomarkers. How sensitive are these to the SNR of the measurements? To the region/volume of interest? To the motion? Do variations in choices impact the conclusions? As the technology matures and new communications arise, emphasizing the rigour of the data analysis, the data analysis pipeline, and the imaging biomarker's robustness becomes more critical. This includes linking the biomarker features to the biophysical interpretation based on histology and what is known of key absorbing structures.

Re: we fully agree with the reviewer that the method for determining the lesion boundary and vessel segmentation is critical to the biomarker computation and the final analysis results. We have added discussion to explain the reason to select the boundaries as *"We selected tissue towards the edge of the lesion to quantify the vascular differences between melanomas and dysplastic nevi for two reasons: 1) to minimize the effects of the melanin signals at the lesion's center and 2) because previous studies have reported that angiogenesis occurs around tumors, altering the vasculature of the tissue bordering a melanoma lesion¹⁻⁵. The edge areas of the scanned melanomas include parts of the tumors' tissue (tumor periphery as confirmed by histology), which exhibits significantly different vascular patterns compared to the adjacent healthy skin, as clearly illustrated in Fig. 3. In addition, dermoscopy (Fig. 3 and Fig. 4) indicated that the pigmentation is generally less dense at the edge of the lesions (tumor periphery) scanned in our study compared to the center of the lesions, affording FRSSOM the highest possible penetration depth within the tumor area (about 1.5 mm at the edge areas as shown in Fig. 3). However, we also found significant differences between the vasculature in the centers of nevi and*

melanoma lesions, despite the higher light attenuation from melanin (supplementary Fig. 4).” Actually the pigmentation of the lesion affects the imaging penetration depth of FRSOM slightly. FRSOM can still resolved the vasculature at the depth of >1.2 mm inside the pigmented lesion and >1.5 mm in the surrounding skin as shown in Fig. 3 and the supplementary videos. We also discuss that multi-spectra FRSOM can separate vascular signals from melanin signals, which allows to directly compute the vascular biomarkers without segmented the boundaries “In order to better visualize and quantify the vasculature under the pigmented areas, multi-spectra or dual-wavelength FRSOM can be used to separate melanin signals from the vascular signals as shown (see Fig. 1 and supplementary Fig. 2), which may allow direct computation of vascular biomarkers without segmenting the boundaries. In the future, we plan to improve FRSOM’s clinical capabilities using the multispectral approach.”

We also have added discussion about the data quality evaluation process:” SNR variations and motion artifacts can significantly affect the determination of the lesion boundary and vessel segmentation, which may degrade the accuracy of the vascular biomarker computation. For all FRSOM measurements, the data quality and SNR of the measurements were pre-screened via a quality control scheme and only high quality datasets were employed in subsequent analyses (see Methods). In the future, the signal quality of recorded FRSOM data can be further evaluated by taking account the skin tone or ulceration of a lesion¹¹ while the motion should be monitored during the whole scanning period to avoid random movements of patients¹². The accuracy of vascular biomarker computation is determined by the boundary and vessel segmentation method. Although our segmentation methods and biomarker computation achieve robust results (see Methods for validation), advanced data processing methods, like deep-learning based approaches¹³, can be applied to directly interpret FRSOM images for diagnosis in the future.”

Variations of SNR could significantly affect the analysis results. However, in our paper, the data quality and SNR of the measurements were pre-screened and only good quality datasets were kept for the analysis, as described in the Methods section. The region/volume of interest was selected to be adjacent to the tumor areas, which is automatically determined by the significant contrast difference between the pigmented lesion and the surrounding skin tissue. The segmentation of the boundary line is quite simple and robust due to this large disparity in contrast. The performance of the segmentation method was validated by two experienced annotators, which showed very high correlation. This has been added as a new subsection (Validation of the boundary and vessel segmentation methods) in the Methods (red text).

Motion has a more significant impact on RSOM image quality and subsequent biomarker computation. With the advancements of FRSOM, data can be acquired in 15 seconds while the patient holds their breath, significantly minimizing motion. In addition, we further applied our previously introduced motion correction methods to reduce the effects of any residual motion. Data with serious motions beyond the correction capability of our motion correction methods were excluded for the analysis as well. The detailed procedure to evaluate the data quality in terms of motion has been described in a new subsection (Data quality control) in the Methods. We also discussed possible ways to further improve the data quality in the discussion.

As the reviewer suggested, we now discuss the importance of the data analysis pipeline and also more advanced data analysis methods for FRSOM image analysis: “Although our segmentation methods and biomarker computation achieve robust results (see Methods for validation), advanced data processing methods, like deep-learning based approaches¹³, can be applied to directly interpret FRSOM images for diagnosis in the future.” For this paper, in order to test the robustness of the existing method, our previous studies have performed validation studies by comparing RSOM images with histology to

check the robustness of our analysis method, which have now been added as a new subsection (Validation of the boundary and vessel segmentation methods) in the Methods (red text).

Reviewer #2 (Remarks to the Author): Expert in melanoma imaging. For the first time, the study uses a fast optoacoustic method (FRSOM) to examine the blood vessels in malignant melanomas in vivo. A small number of 10 melanomas are compared with 10 nevi. The measurements are taken while the patient is holding their breath to minimize motion artifacts. The blood vessels in the skin just outside of the lesion are measured because the melanin in the center covers the vascular signal. Parameters such as vessel density, shape and branching are calculated on the three-dimensional images. The authors found significant differences in the vascular patterns between melanomas and nevi.

1. No information is given on the histological parameters of the lesions. Melanomas are extremely heterogeneous. There are different subtypes. The tumor thickness is an essential parameter for the vessel density and morphology, as could be shown in studies on dynamic OCT of melanomas. Nevi are very heterogenous, too, also in terms of thickness, pigmentation and subtype. It remains unclear whether all of the nevi were dysplastic. Since the thickness and pigmentation of a lesion certainly have a major influence on the representation of the vessels in the FRSOM, the differences found can also only be based on the fact that the melanomas were thicker or more pigmented than the nevi.

Re: As requested by the reviewer, we have now added detailed information on all patients, including the tumor thicknesses and subtypes as determined by histology (supplementary Table II). Three expert dermatologists diagnosed all the nevi we measured as dysplastic and histological analysis confirmed that 8 lesions are atypical junctional nevi and two are compound nevi. As shown in the supplementary Table II, there are no significant differences in the thicknesses and pigmentation between the nevi and melanoma groups, as also confirmed by our expert dermatologists. We fully understand and agree with the reviewer that melanomas and nevi both are extremely heterogeneous and that the different subtypes must be taken into account. This is why we stated in the discussion that *“The results of our study are limited by the small number of subjects. Benign nevi in particular have very diverse features and a larger population is necessary to validate diagnostic performance. A more comprehensive study of a larger number of patients is necessary to validate the computed biomarkers and evaluate how well FRSOM features can distinguish melanoma from benign nevi.”* Although in vivo analysis and larger-scale studies are required to further validate the technique’s capability and feasibility, we believe that the results of our initial pilot study show great potential for FRSOM imaging as a preoperative screening tool, which achieves the AUC value of 0.93 by combining two biomarkers (total blood volume and tortuosity to differentiate melanoma from nevus as shown in Fig. 4.

2. It was not measured in the center of the lesion, but directly outside the melanoma or the nevus in healthy skin: „We selected the vessels in the skin tissue just outside of the lesions (Fig. 4e,j, white dash lines) as regions-of-interest to compare the microvasculature between the nevus and melanoma, since the vasculature optoacoustic signals under the pigmented area may be attenuated by the melanin signals at the wavelength of 532 nm.”

Re: We thank the reviewer for this comment. However, we believe there is a misunderstanding. The vasculature differences are computed in the edge regions of the pigmented lesions as marked in Fig. 3 and Fig. 4 of the paper, which **still lie within the tumor areas (tumor periphery), as confirmed by histology, and not in the healthy skin.** We have improved our description of ROI selection as *“We selected the edge areas of the lesions as regions-of-interest to compare the microvasculature between*

the nevi and melanomas (Fig. 4e,j, white dash lines) because the reduced pigmentation minimizes attenuation due to melanin at the wavelength of 532 nm.” Indeed, while the pigmentation of the lesion could affect the imaging penetration depth, FRSOM could still resolved the vasculature at the depth of >1.2 mm inside the center areas of pigmented lesion and >1.5 mm in the surrounding skin, as shown in Fig. 3 and the supplementary videos. As shown in Fig. 3, the superior imaging performance of FRSOM allowed us to visualize the vasculature in the lesion center, edge, and surrounding skin and resolve significant differences in the vasculature patterns between the edge area and the surrounding healthy skin.

In the center, imaging with vascular signal using RSOM is apparently not possible because the melanin absorbs too much signal. Since the vascular polymorphism correlates with the thickness, it can be assumed that the thick center of a lesion has more conspicuous vessels than the flat macular margin or the healthy skin at the border. The clinical value of the method is therefore very questionable if it cannot be measured at the most pronounced part of the melanoma. Then the higher penetration depth of RSOM compared to dynamic OCT has no added value. The studies in which D-OCT was used to visualize melanoma vessels compared images from the center of the lesion, where the tumor vessels could be visualized despite the lower penetration depth of the OCT signal. Apparently, the vessels within a pigmented lesion cannot be visualized using RSOM, only perilesional vessels. It is not clear why the informative value of this technology should be greater than that of dynamic OCT, which also has a higher resolution. In addition, dynamic OCT has proven itself in everyday clinical practice. An in vivo measurement is possible without any problems on the back or trunk without holding breath, because the measurement is very quick and the handpiece is so light and flexible that it can follow the movements. The only advantage of RSOM over D-OCT, the greater depth of penetration, apparently cannot be used in pigmented lesions.

Re: We thank the reviewer again for the comments. As the reviewer mentioned, *“Since the vascular polymorphism correlates with the thickness, it can be assumed that the thick center of a lesion has more conspicuous vessels than the flat macular margin or the healthy skin at the border.”* As shown in Fig. 3, the vasculature features of melanoma, from the pigmented center to the edge of lesion and to the surrounding skin, are clearly resolved by the FRSOM system, for example FRSOM achieves >1.2 mm penetration depth and >1.5 at the adjacent tissue. We found significantly higher vessel density in the lesion center and edge compared and surrounding healthy skin. The vascular patterns change from the highly dense and disordered pattern in the lesion centre and edge area to a regular vessel network indicative of healthy skin farther from the lesion. These results are similar to the findings of other D-OCT studies, including the study cited by the reviewer.

The reviewer questioned the clinical utility of our method and assumed that FRSOM cannot be used to image pigmented lesions and that there are no technical advantages compared to D-OCT. We respectfully disagree. FRSOM is notably superior to OCT in terms of a) image fidelity due to the so-called “projection artefact of D-OCT” and b) penetration depth. The FRSOM imaging system achieved >1.2 mm penetration depth at the pigmented lesion and >1.5 mm in the adjacent skin tissue, far deeper than possible by D-OCT. Detailed analysis are listed below.

Both advantages arise from the fact that the contrast mechanism of RSOM is direct and based on strong light absorption by blood, while the contrast mechanism of D-OCT is indirect and is based on weak decorrelations caused by blood flow between consecutive images.

In order to show the effects of the projection artefacts, we compared the image quality of normal skin between RSOM and D-OCT (see Fig. 1 below), which has been discussed and clarified in our previous work (Nature Biomedical Engineering, 1(5), 1-8, 2017). We were able to obtain state of the art OCT images from a clinical collaborator (VivoSight Dx from Michelson Diagnostics Ltd., Maidstone, Kent, UK.). It can be noted that the ability of OCT to resolve vessel structures is very limited, especially regarding cross sectional images, highly reducing the depth information provided. The reason is that the projection artefact in the axial direction compromise the ability of optical coherence tomography to obtain cross-sectional images (Fig. 1g) due to the indirect nature of the D-OCT contrast. Therefore, when compared with our RSOM images (Fig. 1a,1c, and 1d), the quality of OCT vascular images are poor (see Fig 1b,e,f).

Figure 1. Comparison of healthy skin images obtained by RSOM and D-OCT. (a) Cross sectional RSOM image of a section of healthy skin (8 mm x 8 mm x 1.3 mm). EP, epidermis; DR, dermis; CL, capillary loops; VP, vascular plexus. (b) Cross sectional D-OCT image of a section of healthy skin (8 mm x 8 mm x 1.3 mm). (c-d) Maximum intensity projections obtained by RSOM in the coronal direction of the capillary loop region and the vascular plexus. (e-f) Maximum intensity projections obtained by D-OCT in the coronal direction of capillary loop region and the vascular plexus.

Regarding penetration depth and the ability to image in the center of the tumor, **we imaged the melanoma lesion in the center, edge, and adjacent skin areas (see Figure 3 and supplementary videos in paper)**. These images exhibited high contrast and revealed different 3D patterns of vasculature in the lesion. Though the imaging penetration depth of FR-SOM can be affected at the center of the lesion due to the strong absorption of melanin, **our FR-SOM imaging system achieved >1.2 mm penetration depth at the center of the pigmented lesion and >1.5 mm in the adjacent skin tissue, much deeper than is possible by D-OCT**. Regarding penetration depth there is a strong consensus in the literature, including the paper cited by the reviewer 2, that indicates that the penetration depth of D-OCT cannot reach beyond ~500 μm when imaging pigmented lesions. We consulted with Dr. Wolfgang Drexler, a leader in the field of OCT, regarding the methods capabilities. He confirmed to us that it is difficult for D-OCT to image below 500 μm depth of pigmented lesions

due to the intrinsic limitations of the D-OCT contrast mechanism. We would be happy to send you a letter from Dr. Drexler supporting this information.

In order to further address the reviewer's concerns, we have computed the vasculature of the lesion center areas between the nevi and melanomas group as shown in Fig. 2 below. All the vascular biomarkers are computed from the FRSOM images acquired from the center pigmented areas of the lesions. We note that the vasculature of the nevus and melanoma in the pigmented areas is clearly resolved with imaging penetration depth of >1.2 mm. The dermal vasculature of the melanoma (Fig. 2h) exhibits a dense and dotted vessel pattern with irregular, dotted, and comma-like structures compared to the nevus (Fig. 2d). Furthermore, we find significant differences in the vascular biomarkers between the nevi (N) and melanomas (M) groups. We have added this Fig. 2 below to the paper as the new supplementary figure 4. These results are similar to studies reported by histological analysis or D-OCT studies as we interpreted in the discussion.

When compared to the results of the vascular biomarkers computed from images acquired in the edge of the lesions (results of Fig. 4 in our paper), these differences between the nevi and melanoma groups in Fig. 2 below are less significant (for example, for the biomarker total blood volume $p=0.001$ vs $p=0.01$). The reason could be that the segmented dermal vessels in the center pigmented areas could be more affected by melanin signals than at the edges when imaging at wavelength of 532 nm. Therefore, we decided to compute the vasculature features of the edge areas in order to minimize the effects of the pigmented signals for the biomarker computation. As shown in our supplementary figure 2, by using dual-wavelength FRSOM, we can actually separate vessels signals from melanin signals, which can minimize the effects of melanin signals for the vasculature biomarker computation and lead better differentiation between nevi and melanomas. We also discussed possible solutions to further improve the performance of FRSOM on pigmented lesion imaging in the discussion section: *"In order to better visualize and quantify the vasculature under the pigmented areas, multi-spectra or dual-wavelength FRSOM can be used to separate melanin signals from the vascular signals as shown (see Fig. 1 and supplementary Fig. 2), which may allow direct computation of vascular biomarkers without segmenting the boundaries. In the future, we plan to improve FRSOM's clinical capabilities using the multispectral approach."*

In summary, FRSOM can resolve the vasculature of the center areas of the pigmented lesions as shown in Fig. 2 with penetration depth of >1.2 mm. We found significant differences in the vasculature in the center areas of the lesion between the nevi and melanomas groups as shown in Fig. 2 below. The imaging performance of FRSOM for melanoma imaging in the pigmented areas can be further improved by multi-spectral FRSOM, which can minimize the effects of melanin signals for the vasculature biomarker computation.

Figure 2. Comparisons of the vasculature features acquired in the pigmented lesion center areas between nevi and melanoma groups. a, Photograph of a dysplastic nevus from a patient chest; the red rectangle indicates the scanning area. b, Cross-sectional MIP image measured at the center areas of the nevus marked by the red rectangle in (a). c,d, Corresponding MIP images in the coronal direction of the epidermis (EP) and dermis (DR) layers of (b). e, Photograph of a melanoma from a patient back. f, Cross-sectional MIP image measured at the center pigmented area of the melanoma marked by the red rectangle in (e). g,h, Corresponding MIP images in the coronal direction of the EP and DR layers of (f); i-n, the computed vessel biomarkers: total blood volume (TBV, i), the vessel density (j), average vessel length (k), tortuosity (l), fractal number (m) and lacunarity (n) between the non-malignant nevi group (N, n=17) and melanoma group (M, n=17). All vessel biomarkers are computed from the MIP FRMOM images of the dermal vessels in the coronal direction; *, and ** represent $P < 0.05$, and $P < 0.01$ respectively. All scale bar: 500 μm .

In addition, we also compared the performance of FRMOM and D-OCT when imaging pigmented lesions, the latter images taken from the literature cited by the second reviewer. As shown in Fig. 3 below, **FRMOM achieved much higher quality images and more than double the penetration depth, even when imaging the center of a pigmented lesion (Fig. 3a-f), compared to D-OCT (limited to less than 500 μm) (Fig. 3g-h).** The vasculature of the pigmented lesions is clearly visualized as shown in the FRMOM images (results of our paper) from the cross sectional and coronal directions (Fig. 3a and 3c, Fig. 3d and 3f). While the work cited by reviewer 2 represents the state-of-the-art in OCT vascular imaging applied to melanoma (see Fig 3g and 3h), the images have much poorer contrast compared to the FRMOM images. While we understand that the comparison is not completely fair since the level of pigmentation for each lesion can be different, it indicates that FRMOM can image deeper than D-OCT which cannot go beyond $\sim 500 \mu\text{m}$ irrespectively of the pigmentation status of the lesion.

Figure 3. (a) Cross sectional FRSOM image of a section of the center of a melanoma lesion (4 mm x 2 mm x 1.3 mm). (b-c) Maximum projection FRSOM images in the coronal direction of the epidermis (EP) and dermis (DR) layers; scale bar 500 μ m. (d) Cross sectional FRSOM image of a section of the center of a dysplastic nevus (4 mm x 2 mm x 1.6 mm). (e-f) Maximum projection FRSOM images in the coronal direction of the epidermis (EP) and dermis (DR) layers; scale bar 500 μ m. (g-h) the vasculature of two melanoma lesions as imaged by D-OCT (6 mm x 6 mm) from the en face view at depth of 300 μ m (Journal of the European Academy of Dermatology and Venereology, 35(5):1087-1093, May 2021).

The supplementary videos of the 3D pigmented nevus and melanoma lesions also demonstrate that FRSOM can visualize the vasculature in the center of pigmented lesions. We selected the edge areas of the lesion to quantify the vasculature differences between melanoma and the dysplastic nevus for two reasons: 1) To minimize the effects of the melanin signals at the lesions' centers; 2) because previous studies have reported that angiogenesis occurs around tumors, altering the vasculature of the tissue bordering a melanoma lesion¹⁻⁵. In addition, the edge area of the melanoma we scanned includes parts of the tumor tissue (tumor periphery confirmed by histology), which exhibits significantly different vascular patterns compared to the adjacent healthy skin as clearly illustrated in Fig. 3 of the paper. In addition, dermoscopy (Fig. 3 and Fig. 4 of the paper) indicated that the lesion edge areas (**tumor periphery**) scanned in our study had generally less dense pigmentation compared to the center areas of the lesions, affording the maximum imaging penetration depth of FRSOM within the tumor (about 1.5 mm at the edge areas as shown in Fig. 3 of the paper). This is actually one of the key findings of our paper. The selected imaging areas were suggested by our expert dermatologists and were afterwards confirmed by histology. In fact, our findings of the patient melanoma studies are similar to previous histological and D-OCT studies which are mentioned in the discussion section, with the key difference that our method achieved much deeper penetration depth and higher contrast of the vasculature.

RSOM has been gradually used in many clinical studies within our groups and our collaborators all over the world [Nature Biomedical Engineering, 1(5), 1-8, 2017, Journal of the American Academy of Dermatology, 84(4), 1121-1123, 2021, The British Journal of Dermatology, 184(2), 352-354, 2020, and more]. Our work further enhances the performance of RSOM to achieve higher speed and better

image quality, which enables for the first time 3D vascular imaging of melanoma at a resolution and imaging depth not possible with other techniques. In addition, the developed FRMOM system is on the way to being transferred to clinical applications and commercialization. Therefore, we believe the clinical utility of our method should have great potential for melanoma imaging.

A formal point of criticism is that the structure of the chapters does not correspond to the content. Results are already presented in the introduction; the methodology is explained in the results section. The chapters need to be sorted better.

Re: To the best of our knowledge, our paper is presented in a manner typical of the field when publishing in a results-before-methods journal such as Nature Communications.

At the end of the introduction we state the core novelties of our paper, without specific results. The numerical values are only the field-of-view and scanning speed, which are necessary to understand the context of the novelties.

At the beginning of the results we provide an overview of the system and image analyses. This is again common for our field in a results-before-methods journal such as Nature Communications.

See for example, Lin, L., Hu, P., Tong, X. et al. High-speed three-dimensional photoacoustic computed tomography for preclinical research and clinical translation. *Nat Commun* 12, 882 (2021). <https://doi.org/10.1038/s41467-021-21232-1>

Significant recent work on blood vessel imaging in melanoma was not cited: Welzel J, Schuh S, De Carvalho N, Themstrup L, Ulrich M, Jemec GBE, Holmes J, Pellacani G. Dynamic optical coherence tomography shows characteristic alterations of blood vessels in malignant melanoma. *J Eur Acad Dermatol Venereol*. 2021 May;35(5):1087-1093. doi: 10.1111/jdv.17080. Epub 2021 Jan 5. PMID: 33300200.

Re: we have included this recent literature in our paper. In addition, we have discussed and compared our findings of the patient melanoma studies with these D-OCT results, which shows similar findings.

1. Marcoval, J., *et al.* Angiogenesis and malignant melanoma. Angiogenesis is related to the development of vertical (tumorigenic) growth phase. *J Cutan Pathol* **24**, 212-218 (1997).
2. Omar, M., Schwarz, M., Soliman, D., Symvoulidis, P. & Ntziachristos, V. Pushing the optical imaging limits of cancer with multi-frequency-band raster-scan optoacoustic mesoscopy (RSOM). *Neoplasia* **17**, 208-214 (2015).
3. Carmeliet, P. & Jain, R.K. Angiogenesis in cancer and other diseases. *Nature* **407**, 249-257 (2000).
4. Forster, J.C., Harriss-Phillips, W.M., Douglass, M.J. & Bezak, E. A review of the development of tumor vasculature and its effects on the tumor microenvironment. *Hypoxia (Auckl)* **5**, 21-32 (2017).
5. Rofstad, E.K., Galappathi, K. & Mathiesen, B.S. Tumor interstitial fluid pressure-a link between tumor hypoxia, microvascular density, and lymph node metastasis. *Neoplasia* **16**, 586-594 (2014).
6. Kashani-Sabet, M., Sagebiel, R.W., Ferreira, C.M., Nosrati, M. & Miller, J.R., 3rd. Tumor vascularity in the prognostic assessment of primary cutaneous melanoma. *J Clin Oncol* **20**, 1826-1831 (2002).

7. Pehamberger, M.B.A.S.U.M.M.H.K.W.H. Quantification of vascularity in nodular melanoma and Spitz's nevus. *J Cutan Pathol*, 272-277 (1997).
8. Rajabi-Estarabadi, A., et al. Optical coherence tomography imaging of melanoma skin cancer. *Lasers Med Sci* **34**, 411-420 (2019).
9. De Carvalho, N., et al. The vascular morphology of melanoma is related to Breslow index: An in vivo study with dynamic optical coherence tomography. *Exp Dermatol* **27**, 1280-1286 (2018).
10. Welzel, J., et al. Dynamic optical coherence tomography shows characteristic alterations of blood vessels in malignant melanoma. *J Eur Acad Dermatol Venereol* **35**, 1087-1093 (2021).
11. Li, X., et al. Optoacoustic Mesoscopy Analysis and Quantitative Estimation of Specific Imaging Metrics in Fitzpatrick skin phototypes II to V. *Journal of biophotonics*, e201800442 (2019).
12. Aguirre, J., et al. Motion quantification and automated correction in clinical RSOM. *IEEE transactions on medical imaging* (2019).
13. Nitkunanantharajah, S., et al. Three-dimensional optoacoustic imaging of nailfold capillaries in systemic sclerosis and its potential for disease differentiation using deep learning. *Scientific reports* **10**, 16444 (2020).

Reviewers' comments:

Reviewer #1 (Remarks to the Author):

The applicants have done well in the Sisyphean task of adequately addressing all comments provided, and the manuscript quality is enhanced.

A couple of the last points the authors may want to address:

a) There still needs to be some sense of scale in figure 2 (a/b). Even though the motion values could be inferred from c & d, it would still be good to provide scale bars and what the axes represent, like in the other figures.

b) The impact of hair on the measurements. Was any preparation done for the 20 lesions before imaging?

c) Perhaps a couple of references can be provided to support the statement:

l 120 "Larger structures are revealed in the lower frequency band and smaller structures in the higher frequency band". While this is known, it would benefit to be supported by primary references.

Reviewer #3 (Remarks to the Author):

Dear Authors,

I read with great interest your study that reports the use of a fast optoacoustic method (FRSOM) to examine the blood vessels in melanomas and nevi in vivo. The study set includes a small subset of lesions, namely 10 melanomas and 10 nevi. FRSOM parameters such as vessel density, shape and branching are calculated on the three-dimensional images. Not surprisingly, significant differences in the vascular patterns between melanomas and nevi were found. It is well known that vessels morphology is strongly related to the malignant or benign nature of the tumor and this has been highlighted with several non invasive imaging devices such as dermoscopy (routinely used in clinical practice), confocal microscopy and OCT.

Thus, the assessment of vessels as a diagnostic feature is not so novel and further it's not clear whether FRSOM can be applied in clinical practice: are those lesions difficult to diagnose based on clinical and dermoscopic analysis? and thus, FRSOM added a new clinical value?

Further to this, there is a lack of in depth description of cases: no data on pigment content (Amelanotic lesion? pigmented?), histologic data (nevi differs in their morphology and melanomas as well-or even more)

Notably, the Authors analyzed the periphery of the lesion and not the center where melanomas are thicker and it's usually the most informative part in terms of diagnostic accuracy.

Besides the value of the study and the paucity of clinical data presented, an additional point is the structure of the paper itself with no clear sections and overlap between them.

Reviewer #1 (Remarks to the Author):

The applicants have done well in the Sisyphean task of adequately addressing all comments provided, and the manuscript quality is enhanced.

A couple of the last points the authors may want to address:

a) There still needs to be some sense of scale in figure 2 (a/b). Even though the motion values could be inferred from c & d, it would still be good to provide scale bars and what the axes represent, like in the other figures.

Re: we thank the reviewer's comment. As the reviewer suggested, we have improved the figure and changes are marked in red color.

b) The impact of hair on the measurements. Was any preparation done for the 20 lesions before imaging?

Re: the hair of the scanned regions has been removed before the measurements. All procedures were performed by our collaborator dermatologists.

c) Perhaps a couple of references can be provided to support the statement:

l 120 "Larger structures are revealed in the lower frequency band and smaller structures in the higher frequency band". While this is known, it would benefit to be supported by primary references.

Re: as the reviewer suggested, we have added two references.

Reviewer #3 (Remarks to the Author):

Dear Authors,

I read with great interest your study that reports the use of a fast optoacoustic method (FRSOM) to examine the blood vessels in melanomas and nevi in vivo. The study set includes a small subset of lesions, namely 10 melanomas and 10 nevi. FRSOM parameters such as vessel density, shape and branching are calculated on the three-dimensional images.

Not surprisingly, significant differences in the vascular patterns between melanomas and nevi were found. It is well known that **vessels morphology is strongly related to the malignant or benign nature of the tumor** and this has been highlighted with several non invasive imaging devices such as dermoscopy (routinely used in clinical practice), confocal microscopy and OCT. Thus, the assessment of vessels as a diagnostic feature is not so novel and further its not clear whether FRSOM can be applied in clinical practice: are those lesions difficult to diagnose based on clinical and dermoscopic analysis? and thus, FRSOM added a new clinical value?

Re: we thank the reviewer for these comments and interest in our work. We also appreciate the effort that it takes to step into a review process to help with its finalization. However, this comment appears to be mixing many aspects of very diverse topics. In decomposing it, we would see that the reviewer:

1. confirms melanoma vasculature as an interesting biomarker,
2. confirms others interest to image this biomarker.

A next logical step here would then be to confirm if our novel technique is superior to previous methods in imaging this target. This is unquestionable, also by the reviewer. For example, a simple inspection of dermoscopy vs. FRSOM reveals that dermoscopy neither sees under the skin nor captures microvascularization:

We note that confocal microscopy can also not visualize microvasculature, as the reviewer notes, unless there are contrast agents used, which is not used in clinical applications.

Therefore, **the clinical value** lies in that we have introduced a technique that images a valuable biomarker with superior imaging performance. The only technique that comes close to the FRMOM performance is (slow) RSOM, which is used as a benchmark in this paper.

For more details on the superiority of FRMOM over all other techniques, we also offer this comprehensive information, now added to the revised paper:

The key purpose of our work is the introduction of new technology (FRMOM) to melanoma vasculature imaging, not a Phase III clinical study to show diagnostic performance. FRMOM imparted scan speeds that were never achieved before, hence leading to the ability to image microvasculature in-vivo in patients for the first time ever.

It is overwhelmingly common in the literature that the first ever introduction of a new technology for clinical interrogation is performed as a pilot study to show efficacy, not a Phase III clinical trial, requiring tens of millions of euros, to prove diagnostic ability.

While from a 30,000 foot perspective view all imaging appears equal, the particular value of the work herein is the specific performance of FRMOM in imaging vasculature. Therefore, in order to dismiss our method it should be shown that the FRMOM performance is inferior or equal to other methods, which however the reviewer never refuted.

Current non-invasive imaging methods cannot adequately resolve microvasculature throughout the entire depth of the skin, which hinders the (future) full exploitation of tumor vascular features as biomarkers. In the introduction of the revised paper, we analytically explain why dermoscopy, confocal microscopy and D-OCT have limited contrast and/or penetration depth to showcase inferior performance to FRMOM.

Neither dermoscopy nor confocal microscopy visualize microvasculature. Compared to D-OCT, that does, as shown in Fig. 1 below,

① FRMOM achieved much higher quality images and more than double the penetration depth, even when imaging the center of a pigmented lesion (Fig. 1a-f), compared to D-OCT (limited to less than 500 μm) (Fig. 1g-h). The OCT images are taken from the literature cited by the second reviewer (Welzel J, et.al., Dynamic optical coherence tomography shows characteristic alterations of blood vessels in malignant melanoma. J Eur Acad Dermatol Venereol. 2021 May.).

② D-OCT does not have the ability to produce detailed cross-sectional images. It only shows en-face images. FRMOM has true three-dimensional imaging power to also produce detailed cross-sections.

③ While the work cited by reviewer 2 represents the state-of-the-art in OCT vascular imaging applied to melanoma (see Fig 1g and 1h), the images have much poorer contrast compared to the FRMOM images. Conversely, the vasculature of the pigmented lesions is clearly visualized by FRMOM from the cross sectional and coronal directions (Fig. 1a and 1c, Fig. 1d and 1f).

These comparisons are not exact, since they are taken from different lesions. Nevertheless, they allow a comparison of FRMOM and D-OCT and confirm what has been already been shown in Aguirre J., et.al. Nat. Biomedical Engineering 2017 and Omar M., et.al. Nature Biomedical Engineering 2019, i.e. that RSOM offers superior performance in microvasculature imaging compared to OCT. In particular, a key novelty in our paper is that FRMOM for the first time enables the microvasculature visualization of human melanoma and adjacent skin tissue on the upper torso at unprecedented resolution and depths up to 1.5 mm, much

deeper than dermoscopy, confocal microscopy, and D-OCT. Furthermore, FRSOM biomarkers calculated for the deeper dermal vasculature quantified obvious morphological differences between the melanoma and benign nevus groups. The comparisons of the six FRSOM biomarkers provides an initial set of vasculature features that are characteristic of melanoma and benign nevi, corresponding well to descriptions from conventional histopathology and OCT studies. Therefore, FRSOM enables rapid, in vivo, noninvasive visualization of the microvascular architecture of melanoma lesions and adjacent skin tissue, which could increase the accuracy of melanoma detection and minimize the need for invasive biopsies. **Such performance was only made possible by the development of this new modality, FRSOM.**

Figure 1. (a) Cross sectional FRSOM image of a section of the center of a melanoma lesion (4 mm x 2 mm x 1.3 mm). (b-c) Maximum projection FRSOM images in the coronal direction of the epidermis (EP) and dermis (DR) layers; scale bar 500 μ m. (d) Cross sectional FRSOM image of a section of the center of a dysplastic nevus (4 mm x 2 mm x 1.6 mm). (e-f) Maximum projection FRSOM images in the coronal direction of the epidermis (EP) and dermis (DR) layers; scale bar 500 μ m. (g-h) the vasculature of two melanoma lesions as imaged by D-OCT (6 mm x 6 mm) from the en face view at depth of 300 μ m (Journal of the European Academy of Dermatology and Venereology, 35(5):1087-1093, May 2021).

All the lesions are diagnosed by our collaborator dermatologists and confirmed with necessary histology analysis. This information is provided in the third paragraph of Method section in the paper.

FRSOM is notably superior to OCT in terms of a) image fidelity due to the so-called “projection artefact of D-OCT” and b) penetration depth. The FRSOM imaging system achieved >1.2 mm penetration depth at the pigmented lesion and >1.5 mm in the adjacent skin tissue, far deeper than possible by D-OCT. In order to demonstrate the performance of FRSOM for melanoma imaging, we have compared D-OCT and FRSOM. This information is also explained in the point-by-point responses to the second reviewer.

Further to this, there is a lack of in depth description of cases: no data on pigment content (Amelanotic lesion? pigmented?), histologic data (nevi differs in their morphology and melanomas as well-or even more)

Detailed information on all lesions in the study are given in supplementary table II, including the subtype of the lesions, the depth, and the location. All lesions are diagnosed and confirmed with histology data, which is provided in the third paragraph of the Method section in the paper. We also addressed a similar comment of the second reviewer, which is given in the responses to the first question of the second reviewer in the point-by-point document.

Notably, the Authors analyzed the periphery of the lesion and not the center where melanomas are thicker and its usually the most informative part in terms of diagnostic accuracy.

Re: There is no imaging method today that can visualize micro-vasculature in the center of melanomas. The reviewer shifts here from comparison to other non-invasive imaging methods, which was the key argument in the comments above, to clinical practice typically achieved with invasive studies using biopsies.

This is an argument that further supports our case. The reviewer supports that imaging deeper is better, and again FR-SOM images deeper than any other non-invasive imaging method, further supporting reasons for clinical value!

We have carefully demonstrated that FR-SOM can image and quantify the vasculature in the center of melanoma lesions. All information has been given in the third and fifth paragraphs of the discussion section in the paper, supplementary figure S4 (quantifications of vasculature in the lesion center), and in the responses to the second question of the second reviewer in the point-by-point document.

In order to further address the reviewer's concerns, we have computed the vasculature of the lesions' center areas between the nevi and melanoma groups, as shown in Fig. 2 below. All the vascular biomarkers are computed from the FR-SOM images acquired from the central pigmented areas of the lesions. We note that the vasculature of the nevus and melanoma in the pigmented areas is clearly resolved with an imaging penetration depth of >1.2 mm. The dermal vasculature of the melanoma (Fig. 2h) exhibits a dense and dotted vessel pattern with irregular structures compared to the nevus (Fig. 2d). Furthermore, we find significant differences in the vascular biomarkers between the nevi (N) and melanoma (M) groups. We have added this Fig. 2 below to the paper as the new supplementary figure 4.

When compared to the results of the vascular biomarkers computed from images acquired in the edge areas of the lesions (results of Fig. 4 in our paper), the vascular differences in the lesions' centers for the nevi and melanoma groups in Fig. 2 below are less significant (for example, for the biomarker total blood volume $p=0.001$ vs $p=0.01$). The reason could be that the segmented dermal vessels in the center pigmented areas could be more affected by melanin signals than at the edges when imaging at wavelength of 532 nm. Therefore, we decided to compute the vasculature features of the edge areas in order to minimize the effects of the pigmented signals for the biomarker computation. As shown in our supplementary figure 2, by using dual-wavelength FR-SOM, we can actually separate vessels signals from

melanin signals, which can minimize the effects of melanin signals for the vasculature biomarker computation and lead better differentiation between nevi and melanomas. We also discussed possible solutions to further improve the performance of FRsOM on pigmented lesion imaging in the discussion section: “In order to better visualize and quantify the vasculature under the pigmented areas, multi-spectra or dual-wavelength FRsOM can be used to separate melanin signals from the vascular signals as shown (see Fig. 1 and supplementary Fig. 2), which may allow direct computation of vascular biomarkers without segmenting the boundaries. In the future, we plan to improve FRsOM’s clinical capabilities using the multispectral approach.”

In summary, FRsOM can resolve the vasculature of the center areas of the pigmented lesions as shown in Fig. 2 with penetration depth of >1.2 mm. We found significant differences in the vasculature in the center areas of the lesion between the nevi and melanomas groups as shown in Fig. 2 below. The imaging performance of FRsOM for melanoma imaging in the pigmented areas can be further improved by multi-spectral FRsOM, which can minimize the effects of melanin signals for the vasculature biomarker computation.

Figure 2. Comparisons of the vasculature features acquired in the pigmented lesion center areas between nevi and melanoma groups. a, Photograph of a dysplastic nevus from a patient chest; the red rectangle indicates the scanning area. b, Cross-sectional MIP image measured at the center areas of the nevus marked by the red rectangle in (a). c,d, Corresponding MIP images in the coronal direction of the epidermis (EP) and dermis (DR) layers of (b). e, Photograph of a melanoma from a patient back. f, Cross-sectional MIP image measured at the center pigmented area of the melanoma marked by the red rectangle in (e). g,h, Corresponding MIP images in the coronal direction of the EP and DR layers of (f); i-n, the computed vessel biomarkers: total blood volume (TBV, i), the vessel density (j), average vessel length (k), tortuosity (l), fractal number (m) and lacunarity (n) between the non-malignant nevi group

(N, n=17) and melanoma group (M, n=17). All vessel biomarkers are computed from the MIP FRSOM images of the dermal vessels in the coronal direction; *, and ** represent $P < 0.05$, and $P < 0.01$ respectively. All scale bar: 500 μm .

The supplementary videos of the 3D pigmented nevus and melanoma lesions also demonstrate that FRSOM can visualize the vasculature in the center of pigmented lesions. We selected the edge areas of the lesion to quantify the vasculature differences between melanoma and the dysplastic nevus for two reasons: 1) To minimize the effects of the melanin signals at the lesions' centers; 2) because previous studies have reported that angiogenesis occurs around tumors, altering the vasculature of the tissue bordering a melanoma lesion[1-5]. In addition, the edge area of the melanoma we scanned includes parts of the tumor tissue (tumor periphery confirmed by histology), which exhibits significantly different vascular patterns compared to the adjacent healthy skin as clearly illustrated in Fig. 3 of the paper. In addition, dermoscopy (Fig. 3 and Fig. 4 of the paper) indicated that the lesion edge areas (tumor periphery) scanned in our study had generally less dense pigmentation compared to the center areas of the lesions, affording the maximum imaging penetration depth of FRSOM within the tumor (about 1.5 mm at the edge areas as shown in Fig. 3 of the paper). This is actually one of the key findings of our paper. The selected imaging areas were suggested by our expert dermatologists and were afterwards confirmed by histology. In fact, our findings of the patient melanoma studies are similar to previous histological and D-OCT studies which are mentioned in the discussion section, with the key difference that our method achieved much deeper penetration depth and higher contrast of the vasculature.

Besides the value of the study and the paucity of clinical data presented, an additional point is the structure of the paper itself with no clear sections and overlap between them.

Re: While we have improved further now the structure of the paper, we note that Reviewer #1 has the opposite opinion. In any case we can further improve the structure if the reviewer provides specific examples where the structure is inadequate.

To the best of our knowledge, however, our paper is presented in a manner typical of the field when publishing in a results-before-methods journal such as Nature Communications.

At the end of the introduction, we state the core novelties of our paper, without specific results. The numerical values are only the field-of-view and scanning speed, which are necessary to understand the context of the novelties.

At the beginning of the results, we provide an overview of the system and image analyses. This is again common for our field in a results-before-methods journal such as Nature Communications.

See for example, Lin, L., Hu, P., Tong, X. et al. High-speed three-dimensional photoacoustic computed tomography for preclinical research and clinical translation. *Nat Commun* 12, 882 (2021). <https://doi.org/10.1038/s41467-021-21232-1>

- [1] J. Marcoval *et al.*, "Angiogenesis and malignant melanoma. Angiogenesis is related to the development of vertical (tumorigenic) growth phase," (in English), *J Cutan Pathol*, vol. 24, no. 4, pp. 212-218, Apr 1997, doi: DOI 10.1111/j.1600-0560.1997.tb01583.x.

- [2] M. Omar, M. Schwarz, D. Soliman, P. Symvoulidis, and V. Ntziachristos, "Pushing the optical imaging limits of cancer with multi-frequency-band raster-scan optoacoustic mesoscopy (RSOM)," *Neoplasia*, vol. 17, no. 2, pp. 208-14, Feb 2015, doi: 10.1016/j.neo.2014.12.010.
- [3] P. Carmeliet and R. K. Jain, "Angiogenesis in cancer and other diseases," *Nature*, vol. 407, no. 6801, pp. 249-57, Sep 14 2000, doi: 10.1038/35025220.
- [4] J. C. Forster, W. M. Harriss-Phillips, M. J. Douglass, and E. Bezak, "A review of the development of tumor vasculature and its effects on the tumor microenvironment," *Hypoxia (Auckl)*, vol. 5, pp. 21-32, 2017, doi: 10.2147/HP.S133231.
- [5] E. K. Rofstad, K. Galappathi, and B. S. Mathiesen, "Tumor interstitial fluid pressure-a link between tumor hypoxia, microvascular density, and lymph node metastasis," *Neoplasia*, vol. 16, no. 7, pp. 586-94, Jul 2014, doi: 10.1016/j.neo.2014.07.003.

REVIEWERS' COMMENTS

Reviewer #4 (Remarks to the Author):

The authors addressed almost all points raised by the reviewers. However, the question on the clinical utility and applicability of the method remains to be determined.

Questions:

a) The term "junctional nevus" refers to an intra-epidermal proliferation - thus it is not clear how the authors calculate the level of thickness in the nevi listed in the supplementary file.

b) Could age, location and/or presence of sun-damage in melanoma patients have an impact on the observed vascularity at the periphery of the lesion? Are there any data showing that skin vascularity is not affected by age, skin thinning or location.

REVIEWERS' COMMENTS

Reviewer #4 (Remarks to the Author):

The authors addressed almost all points raised by the reviewers. However, the question on the clinical utility and applicability of the method remains to be determined.

Re: We thank the reviewer for the comments and agree that the clinical utility and applicability of the method will require further validations for future clinical translation. This is a pilot study, which introduces a novel technology for melanoma vasculature imaging. We have discussed this in the discussion of the paper: *"The results of our study are limited by the small number of subjects. Benign nevi in particular have very diverse features and a larger population is necessary to validate diagnostic performance. A more comprehensive study of a larger number of patients is necessary to validate the computed biomarkers and evaluate how well FRSOM features can distinguish melanoma from benign nevi."* *"In order to better visualize and quantify the vasculature under the pigmented areas, multi-spectra or dual-wavelength FRSOM can be used to separate melanin signals from the vascular signals as shown (see Fig. 1 and supplementary Fig. 2), which may allow direct computation of vascular biomarkers without segmenting the boundaries. In the future, we plan to improve FRSOM's clinical capabilities using the multispectral approach."*

Questions:

a) The term "junctional nevus" refers to an intra-epidermal proliferation - thus it is not clear how the auroras calculate the level of thickness in the nevi listed in the supplementary file.

Re: according to our pathologist who was responsible for determining the thickness of lesions, the thickness of the nevi was always calculated from the directly overlaying stratum granulosum to the deepest point of the nevus, e.g, to the basal point of a nest. The measurements were carried out according to the Breslow Index in Melanoma. We have added this information marked in red color in the Method section: *"According to the Breslow Index in Melanoma, the thickness of the nevi was calculated from the directly overlaying stratum granulosum to the deepest point of the nevus, e.g, to the basal point of a nest."*

b) Could age, location and/or presence of sun-damage in melanoma patients have an impact on the observed vascularity at the periphery of the lesion? Are there any data showing that skin vascularity is not affected by age, skin thinning or location.

Re: Literature shows that there are possible variations of healthy skin microvasculature due to the age, location, or presence of sun-damage, but few studies investigate these effects on the vascularity of melanoma patients. However, we believe that these factors will not significantly affect our results because vascular changes caused by melanoma are more significant compared to those expected from aging, location or sun-damage. For example, it was reported that there is marked loss in the cutaneous microvasculature density for both the chronologic aging and photoaging compared to young volunteers

(Kelly et al. The Effect of Aging on the Cutaneous Microvasculature, 1995). In our work, the melanoma patient group had an average age of 70.0 ± 14.6 years while the nevus group had an average of 48.5 ± 18.6 years. However, our results show that the melanoma group has significantly higher vascularity compared to the nevi group. This suggests that chronologic aging and photoaging are not important factors to affect the vascularity of the lesion. Regarding the location, the nevi and the melanoma groups were measured at various locations on the body and we do not observe bias due to these body locations in the two groups.

We added this information in the Discussion section marked in red color *“The increased vascularity in the melanoma group also runs counter to expected reductions in skin microvasculature due to aging and photoaging [50], since our melanoma group had a significantly higher mean age compared to the nevi group, suggesting that these factors do not appreciably affect our results.”*